# Meta-analysis-derived estimates of stressor–response associations for riverine organism groups

Freshwater ecosystems, particularly rivers, are experiencing the most rapid biodiversity declines of any biome, driven by several interacting stressors operating across local to global scales. Despite growing research on these interactions, the lack of systematic quantification of individual stressor gradients limits our ability to disentangle their cumulative effects. Here we present a global synthesis of stressor–response relationships across five key riverine organism groups—prokaryotes, algae, macrophytes, invertebrates and fish. We screened 22,120 papers and extracted 276 studies with 1,332 stressor–response relationships. We used generalized linear mixed models and Bayesian meta-analyses to quantify the response to the seven most prevalent stressors. Consistently across taxa, biodiversity loss (taxon richness and evenness) reflected elevated salinity, oxygen depletion and fine sediment accumulation, while the association with nutrient enrichment and warming varied among groups. Predictive tools, including hypothetical outcome plots and partial dependence plots, revealed the interplay of stressors and predicted biodiversity response to stress increase. Our findings establish a quantitative baseline for a continuous global synthesis, refining predictions of anthropogenic stressor impacts, identifying key research gaps and informing conservation strategies for freshwater ecosystems.

Freshwater ecosystems—particularly rivers—have undergone rapid biodiversity declines in recent decades, driven by several interacting stressors across local, catchment and global scales[1,2]. Agricultural intensification, urban wastewater and sewer overflows degrade water quality[3–5], while water abstraction exacerbates droughts and impervious surfaces intensify flash floods[6]. Other catchment-scale pressures, such as land reclamation, hydropower generation and navigation, further degrade habitat structure[7,8]. Meanwhile, global change further intensifies these impacts by disrupting flow regimes and thermal dynamics, compounding local pressures[9,10]. All these stressors may interact in complex ways, shaping the composition and diversity of riverine communities and making it challenging to predict biodiversity responses. Understanding these relations and the underlying mechanisms is crucial for effective conservation and management.

Over the past decade, research has increasingly focused on the cumulative effects of several stressors on riverine biodiversity. For example, Lemm et al.[11] found that the ecological status of European rivers declines with increasing intensity of several stressors, while Brauns et al.[12] demonstrated how several stressors impair ecosystem functioning. Experimental studies have explored how stressors interact—whether their effects are additive, synergistic or antagonistic—and investigated the underlying mechanisms[13–15]. However, despite substantial progress, a generalizable framework for predicting how several stressors collectively shape biodiversity remains elusive. This would require information, ideally raw data, on the relation between various organism groups and stressors from a range of ecoregions and river types, and a model that accounts for these multivariable data and the associated bias.

Paradoxically, the focus on multistressor impacts has obscured a critical knowledge gap: the absolute effects of individual stressors

✉e-mail: willem.kaijser@uni-due.de; christian.schuerings@uni-due.de; daniel.hering@uni-due.de

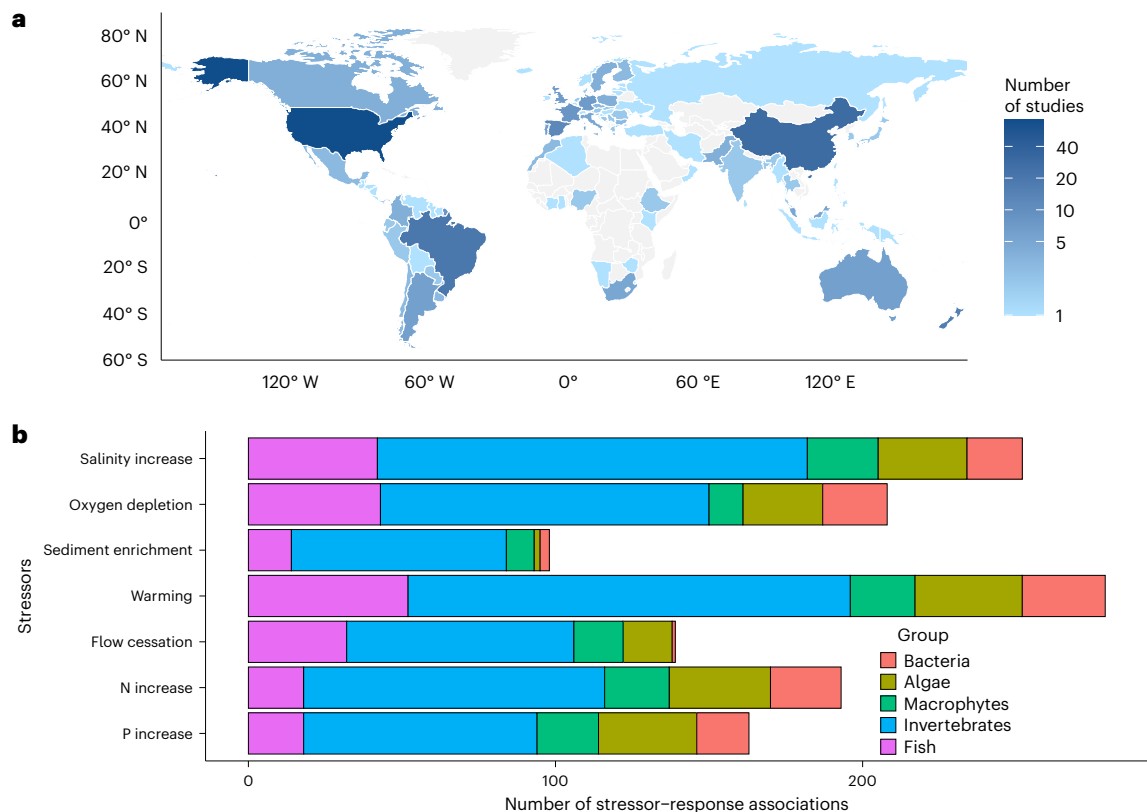

**Fig. 1 | Data sources. a,b,** Geographic distribution of data sources included in this synthesis (**a**) and breakdown of stressor categories by organism groups (**b**). Map data from Natural Earth (http://www.naturalearthdata.com).

and their interactions with specific organism groups remain poorly quantified. Although numerous studies have examined individual stressor–response relationships[15] and one global synthesis compared terrestrial, marine and freshwater communities[16], a quantitative assessment across aquatic organism groups and stressors is still lacking.

Stressors affecting aquatic organisms can be broadly categorized as physicochemical stressors, which alter water quality and hydromorphological stressors, which modify habitat structure. Each stressor operates through distinct modes of action that may be caused by specific cellular mechanisms or through the provision/removal of habitats and that exclude or favour certain species. For instance, oxygen depletion slows metabolism[17,18], while elevated salinity disrupts osmoregulation[19]. Hydromorphological changes, such as fine sediment accumulation or channelization, alter habitat availability, favouring some species while excluding others[20].

The sensitivity of riverine organisms to these stressors varies widely. Larger organisms, such as fish and macrophytes, are disproportionately affected by habitat modifications including associated dispersal constraints[21,22], while physicochemical stressors such as oxygen depletion and warming can affect a broader range of taxa[23,24]. Yet, systematic comparisons of stressor associations across different taxonomic groups remain rare[20,22], limiting our ability to extrapolate localized findings to broader ecological contexts.

To address this gap, we present a global synthesis of stressor–response relationships across five key riverine organism groups—bacteria/archaea, algae, macrophytes, invertebrates and fish—focusing on seven widespread stressors. Drawing from 22,120 observational studies, we compiled 276 datasets encompassing 1,332 distinct stressor–response relationships. Each study sampled a riverine community at least six times (median = 14, mean = 58, s.d. = 346) focusing on at least one stressor (median = 3, mean = 3.3, s.d. = 2.5). We did not include experimental studies, as our prime interest lies in the relationship

between stressors and biota under real-world conditions. Using multivariable generalized linear mixed models (GLMMs) and Bayesian meta-analyses, we quantified the associations of individual stressors and identified overarching response patterns.

This analysis provides a quantitative baseline for a continuous, global assessment of freshwater stressor impacts, enhancing our ability to predict biodiversity responses under increasing anthropogenic pressures. The prime objective was to establish empirical relationships between stressor intensity and biodiversity metrics over a wide range of conditions, independently of possible causes. Our findings offer crucial insights for conservation planning, informing mitigation strategies tailored to specific stressors and organism groups. By systematically quantifying stressor–response relationships, we contribute to a more predictive and actionable understanding of riverine ecosystem resilience in the face of accelerating environmental change.

## Results and discussion

### A global perspective on stressor–response relationships

Our meta-analysis revealed 1,332 stressor–response relationships from 276 studies across 87 countries (Fig. 1).

Nearly half the identified stressor–response relationships focused on invertebrates, with salinity and temperature being the most frequently studied stressors. In contrast, hydromorphological stressors—despite their recognized ecological importance[11]—were under-represented (Fig. 1), underscoring a critical research gap.

Taxonomic richness (for example, species or genus counts) that were analysed with log-linear models prevailed (92.9%), while for all logit-linear models evenness prevailed (71%) (Supplementary Table 4). In many individual studies, the relationships between stressors and biological responses varied greatly, often showing substantial heterogeneity. Although some stressor–response patterns are evident, the overall relations were relatively modest and highly variable for

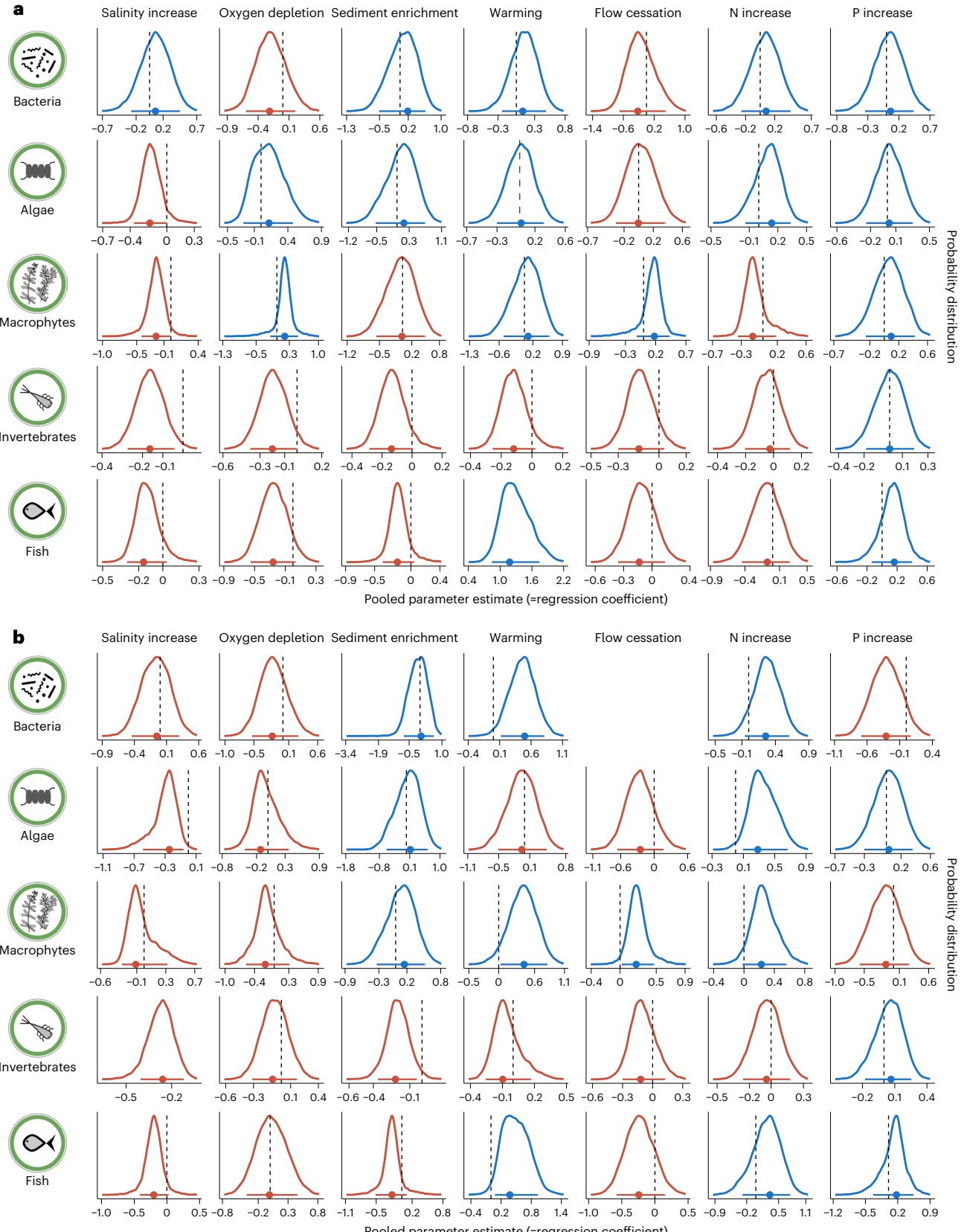

**Fig. 2 | Stressor–response relationships.** Posterior probability distributions (approximately the same as frequency of simulations) of regression coefficients for stressor–response relationships across five organism groups and seven stressors. **a**,**b**, Regression coefficients from log-linear models for taxonomic richness (**a**) and logit-linear models for evenness/coverage (**b**). Dots represent the mode, known as the MAP estimate; horizontal bars indicate 90% high-density intervals. Blue curves indicate positive mode estimates (approximately the same as positive response of organism group to stressor), while red curves indicate negative estimates. Full posterior results are given in Supplementary Data 2.

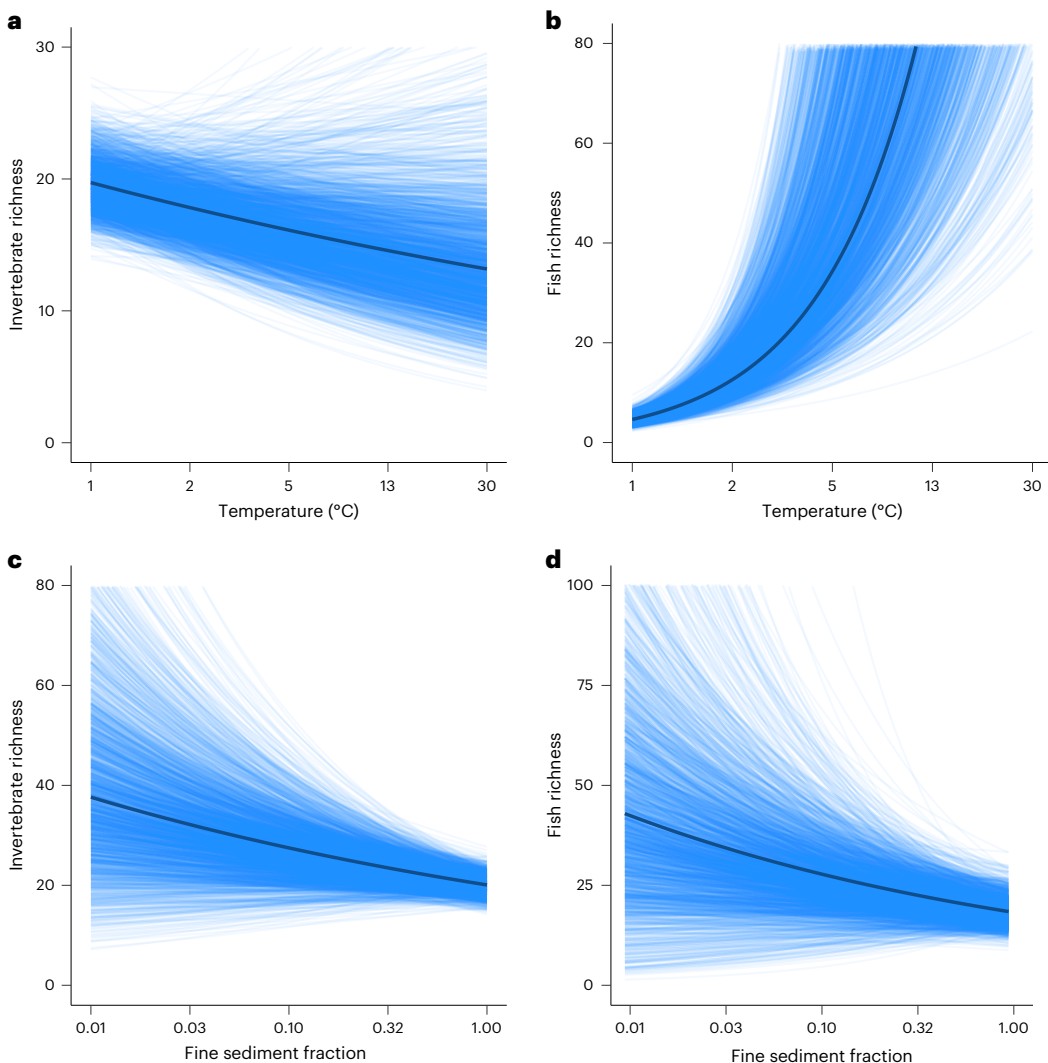

**Fig. 3 | Hypothetical outcome plots. a–d**, The HOPs predict taxonomic richness associations between invertebrates and temperature (**a**), fish and temperature (**b**), invertebrates and fine sediment (**c**) and fish and fine sediment (**d**).

different stressors and organism groups. This highlights the inherent complexity of interpreting stressor–response dynamics in ecological systems (compare Supplementary Data 2).

### Relations of biodiversity patterns to single and combined stressor gradients

Only invertebrates consistently showed strong and negative relations to all stressors, except phosphorus enrichment, reflecting their dependence on oxygen availability, habitat structure and stable flow conditions[25–27]. In contrast to invertebrates, microorganisms—particularly bacteria/archaea—showed more variable patterns, probably due to their dependence on microscale conditions and the limited availability of suitable datasets[28–30]. Fish exhibited a mix of positive and negative stressor–response relationships, while relations of macrophytes to oxygen depletion and flow is opposing those of other groups, emphasizing their unique adaptations as sessile autotrophs (Fig. 2). However, our meta-analysis necessarily obscures divergent patterns of macrophytic taxa, such as bryophytes and vascular plants, which vary substantially in evolutionary origins, traits and environmental requirements.

Microbial responses require a dedicated sampling strategy to reflect microscale patterns. The response of bacteria/archaea was often divergent compared with macroorganisms, probably due to their environmental specificity and the under-representation of relevant

studies. This includes methodological challenges to designate species and to assign operational taxonomic units that respond to stressor intensities. Notably, bacterial diversity exhibited a positive relationship with temperature, consistent with findings in marine systems[31–33]. Nutrient enrichment showed contrasting relations: bacterial diversity decreased with phosphorus—aligning with global patterns observed in lakes and streams[34]—but increased with nitrogen, probably due to the direct dependency of many taxa on nitrogen resources. Negative associations were frequently observed at higher nutrient concentrations, in the order of magnitude of 10 mg l$^{-1}$ total nitrogen[35], while the mean of all bacteria studies analysed here was 2.17 mg l$^{-1}$ (s.d. = 0.8). Salinity and oxygen depletion showed no clear trends.

Algae, covering both planktonic and benthic algae, showed strong curve-linear relations to salinity and nutrient levels. The negative relationship to salinity was particularly strong, probably driven by osmotic stress[19,36], supporting the general accepted conjecture that freshwater species are not directly replaced by brackish water species when salinity increases. Algal richness and evenness were positively associated with nutrient enrichment, particularly nitrogen, reflecting enhanced productivity[37]. This result contrasts with the general assumption that eutrophication is a prime stressor to phytoplankton biodiversity[35]—obviously, moderate nutrient input can enable higher alpha-diversity of algae. However, the more frequent

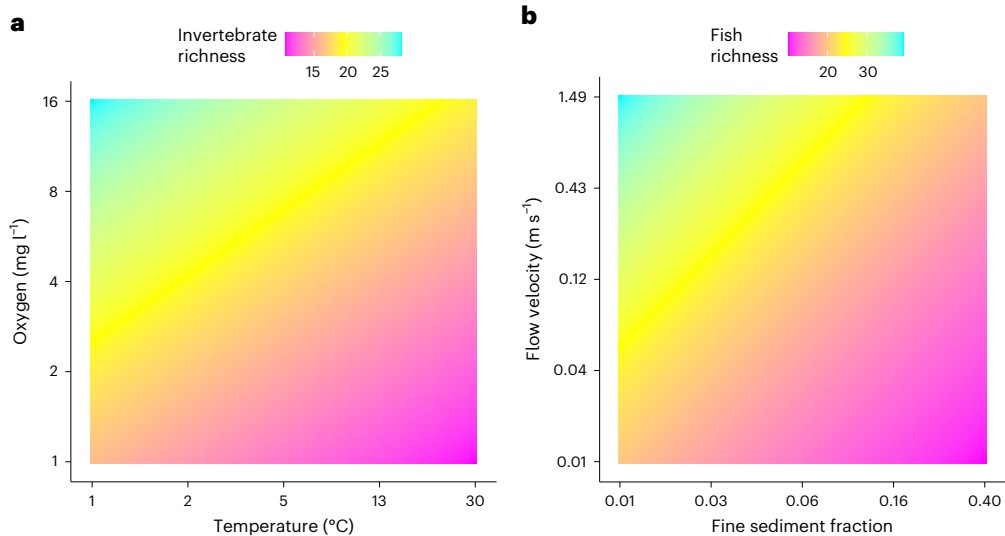

**Fig. 4 | Partial dependence plots. a,b,** The PDPs illustrate invertebrate richness (number of taxa) as a function of temperature and oxygen concentration (**a**) and fish richness (number of taxa) as a function of fine sediment enrichment and flow velocity (**b**), assuming additive stressor relationships.

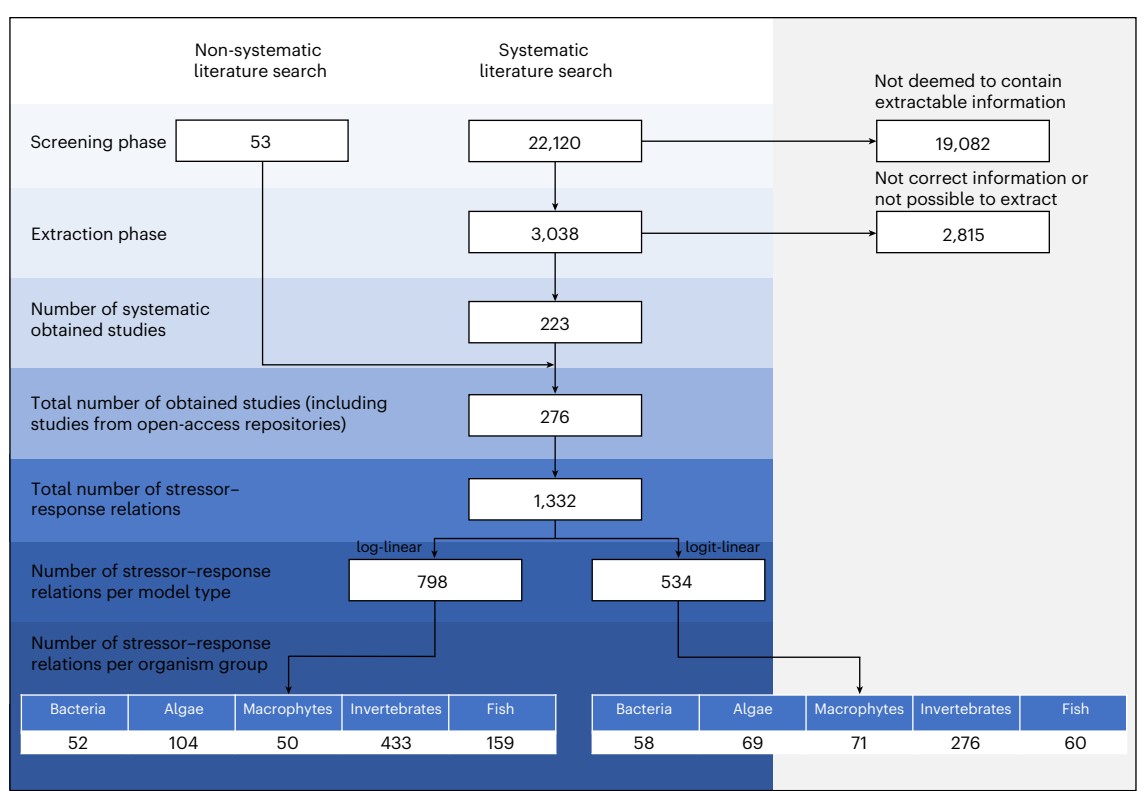

**Fig. 5 | Article selection procedure.** Flowchat depicting how the articles used in the meta-analysis were identified.

occurrence of generalists and competitive species under eutrophic conditions may decline beta- and gamma-diversity because of disappearance of specialists for oligotrophic conditions[38]. The observed positive relationship may simply reflect higher specimen numbers in eutrophic waters, which increase the likelihood of detecting more species. Other stressor–response relations were weaker, suggesting indirect relations (for example, through another environmental variables) and data insufficiency.

Macrophytes (aquatic bryophytes and vascular plants) showed unique stressor–response patterns. They were positively associated with oxygen and flow cessation, probably benefiting from the stabilization of sediments and reduced force of current[39,40]. However, they showed negative relationships with salinity and nitrogen enrichment, which might favour a limited number of tolerant and competitive species outcompeting others and thus reducing species number and evenness[41,42]. These distinct associations highlight the specialized ecological niches occupied by macrophytes in river systems.

Invertebrates exhibited predominantly negative associations with both habitat and water quality stressors, particularly salinity, oxygen depletion, fine sediment accumulation and warming. These patterns probably reflect their sensitivity to habitat degradation and declining water quality[18,25,43]. Nutrient enrichment was only weakly associated,

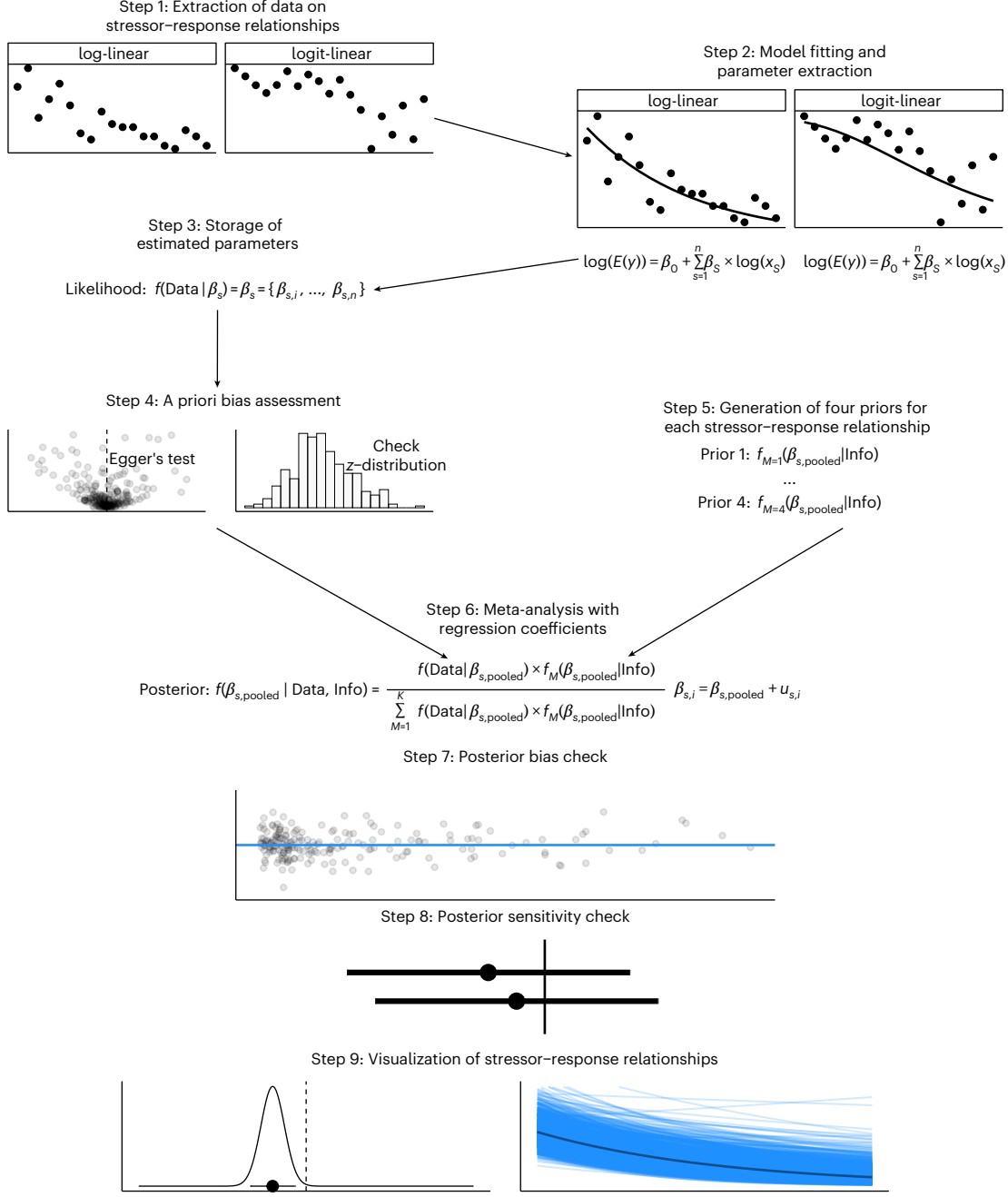

**Fig. 6 | Analytical workflow.** Steps for data extraction, model fitting, parameter storage, bias assessment, prior generation and meta-analysis.

suggesting that direct nutrient impacts may be less relevant than habitat alterations and acts in an indirect way, for example through temporary oxygen depletion[20].

Fish exhibit a mix of positive and negative stressor–response relationships. Negative relations were observed for salinity, oxygen depletion, sediment accumulation and flow cessation, reflecting habitat degradation and metabolic constraints[44]. Conversely, positive relationships with rising temperatures may reflect the dominance of warm-water species in downstream fish regions, which naturally exhibit a higher species richness compared with the cooler upstream areas; however, other factors such as habitat characteristics shape these patterns[45,46]. Nutrient enrichment showed minimal patterns and probably influenced fish only indirectly via oxygen depletion, habitat structure or prey availability.

Hypothetical outcome plots (HOPs) (Fig. 3) illustrate associations between taxonomic richness and selected key stressors, chosen on the basis of strength and uncertainty of observed relationships. Invertebrate and fish richness showed contrasting associations to increasing temperature (Fig. 3a,b), with a decline in invertebrate richness and a positive trend for fish richness. These patterns reflect differences in thermal sensitivity and physiological constraints. Invertebrate richness declined with warming, probably reflecting reduced oxygen availability. In contrast, fish richness tended to increase with moderate temperature rises, potentially coinciding with expanding river size naturally positively correlated with fish richness[47] seasonal patterns[48], as well as increase in exotic species[49]. Both, invertebrate and fish richness were negatively related to fine sediment accumulation and flow cessation, respectively (Fig. 3c,d), emphasizing the negative

association between excessive sedimentation and reduced flow velocity on habitat complexity and potentially on oxygen availability. Fine sediment accumulation has been recognized as a key threat for riverine macroinvertebrates[50,51] and fish[52], but quantifications of stressor–responses relations remain scarce.

Partial dependence plots (PDPs) (Fig. 4) illustrate the additive relation of two key stressors on taxonomic richness, highlighting how biodiversity related to several environmental stressors. Invertebrate richness as a function of temperature and oxygen depletion revealed that while high oxygen levels promote taxa richness, these benefits are reduced by elevated temperatures (Fig. 4a). Fish richness was consistently lower with higher fine sediment fraction and lower flow velocity (Fig. 4b), two hydromorphological stressors that are known to jointly reduce habitat quality and complexity. These findings reinforce the need for holistic management approaches, as the effectiveness of mitigating a single stressor may depend on the presence or severity of others that account for multistressor conditions rather than addressing stressors in isolation.

## Management and research implications

Our results emphasize the importance of management strategies tailored to regional conditions and stressor intensities[53]. Consistently, all taxa were strongly and negatively associated with salinity. As salinity is strongly correlated with other stressors, such as pesticides[54], it may serve as a proxy for broader ecological degradation. Other stressors showed more variable associations. For example, warming was positively associated with the diversity of most organism groups, in particular fish, but negatively with invertebrates. At least in parts, this observation might simply reflect higher species numbers of warmer, downstream river reaches, which is most obvious for fish, while species richness of invertebrates typically declines downstream. These findings highlight the limitations of simplistic biodiversity metrics, such as species richness, which may obscure declines in specialist or functionally important taxa due to the proliferation of generalists. Despite these complexities, several overarching patterns emerged: salinization, oxygen depletion, fine sediment accumulation and flow cessation consistently related negatively with biodiversity, particularly for fish and invertebrates. In contrast, nutrient enrichment had minimal relations to most taxa but was strongly positively associated to algal diversity, although this observation might have been caused by the higher specimen numbers in samples from eutrophic waters potentially leading to reduced beta-diversity[38].

Future research should aim to disentangle these relationships, identifying primary drivers to improve mitigation strategies. Addressing the observed variability in stressor–response patterns will require integrating local environmental conditions, species-specific tolerances and multivariable stressor interactions into ecological assessments.

Effective management of rivers must prioritize pollution reduction, sediment budget restoration and improved flow regimes to safeguard biodiversity. Additionally, fostering transparent data sharing and integrative modelling approaches will enhance our ability to refine stressor–response relationships and advance ecological understanding. By building on the findings of this study, future research can drive information-based conservation actions and support the sustainable management of aquatic ecosystems.

Beyond estimating the stressor–response relationships, this synthesis highlights the need for improved ecological data reporting. The available data reflect the research priorities of recent decades rather than the actual relevance of stressors, organism groups, or river types. The dataset is heavily dominated by studies from a few countries—particularly the USA, China and Germany—and macroinvertebrates are substantially better represented than other organism groups. In addition, several emergent stressors, such as contaminants and invasive species, were not considered because of difficulties in parameterization. In addition, small sample sizes and incomplete reporting constrained our ability to account for spatial and temporal autocorrelation, probably contributing to the heterogeneity among stressor–response relationships. Finally, the reliance on summary statistics (for example, means, medians and standard deviations) and the lack of accessible raw datasets constrain the precision of meta-analyses, a cumulative scientific approach, and hinder the detection of ecological patterns relevant to conservation and management. Greater availability of multivariate datasets, coupled with data-sharing practices, would enhance the transferability of findings. The dataset and analytical framework presented here provide a foundation for future assessments, offering an adaptable structure that can be updated using the accompanying code or by incorporating posterior estimates as priors in subsequent analyses. This flexibility supports ongoing monitoring efforts and ensures that findings remain relevant for guiding ecological research and management.

Bayesian approaches offer a promising avenue for synthesizing and integrating diverse data sources while accounting for parameter uncertainties[55]. Incorporating methods such as Bayesian network meta-analyses[56] enables estimation of missing components by leveraging existing models, facilitating more robust predictions of stressor impacts. The compiled data source will act as a continuously improving baseline for the quantification of stressor–response relationships. Expanding predictive modelling capabilities through Bayesian methods can further refine stressor–response assessments, improving management strategies. These models can inform predictive-scenario testing, enabling managers to evaluate alternative stressor reduction strategies and their expected outcomes on biodiversity. Integrating such approaches into conservation planning will enhance our ability to mitigate anthropogenic stressors and strengthen freshwater ecosystem resilience.

## Methods

### Overview

We systematically identified studies that examined stressor–response relationships under field conditions. For each study, we extracted the data and fitted separate GLMMs to each dataset. The resulting parameter estimates (regression coefficients) from these individual models were then synthesized through a meta-analysis using Bayesian model averaging.

### Step 1: Data collection and literature search

We compiled a comprehensive dataset on stressor–response relationships in riverine ecosystems under field conditions through a systematic Web of Science search. We encompassed five key riverine organism groups—bacteria/archaea, algae, macrophytes, invertebrates and fish—and seven major anthropogenic stressors—salinity increase, oxygen depletion, fine sediment enrichment, temperature increase, flow cessation and nitrogen and phosphorus enrichment. To ensure ecological realism, we excluded laboratory-based experiments, restricting our analysis to field studies. This search initially yielded 22,120 articles, which were systematically screened, resulting in 223 retained studies (Supplementary Information (step 1)). We supplemented this dataset with 55 additional studies from the open-access repositories Dryad, GitHub and Zotero, yielding a final dataset of 276 studies[22,25,57–330] encompassing 1,332 quantified biota–stressor relationships (Fig. 5).

We extracted data on key response variables, including taxonomic richness and evenness from figures, tables and supplementary datasets, prioritizing the most commonly reported metrics for each organism group. As independent variables, we extracted proxies of stressor intensities (for a full list of proxies compare Supplementary Information (step 1)). We also extracted if the study was based on a temporal or a spatial gradient (Fig. 6 (step 1) and Supplementary Information (step 1)). We refer to the data extracted from an individual study as 'individual dataset' in the following.

### Step 2: Model fitting and parameter extraction

The purpose of this step was to derive the model parameters and standard errors of the GLMM between each stressor and biotic response for each individual dataset, that is for each stressor–response relationship stemming from a single study. We applied GLMMs to model the response variables as follows. The linear component of the GLMMs was modelled with a log-link (taxonomic richness; count data) or with the logit-link (evenness and coverage; proportional data) (Fig. 6 (step 2)).

To facilitate parameter estimation, all independent variables were log-transformed (natural log), ensuring that model parameters corresponded to elasticity and semi-elasticity coefficients[331]. Elasticity coefficients, estimated via log-linear models, quantify the percentage change in a response variable per 1% increase in stressor intensity (for example, an elasticity of 0.2 indicates a 0.2% response change). Semi-elasticity coefficients, estimated via logit-linear models, measure the change in the logged odds of the response variable per 1% change in stressor intensity. When approaching zero, they closely approximate elasticity coefficients, enhancing interpretability.

This approach enables three key advantages: (1) the use of general priors across models, (2) comparable interpretation of stressor impacts and (3) the avoidance of self-referential issues and of bias introduced by z-transformations or minimum-maximum transformations, facilitating comparisons across models (Supplementary Information (step 2)).

If studies provided information on sampling dates, seasons or years, or on individual rivers sampled, random effects were applied if they did not lead to convergence issues in the model.

### Step 3: Storage of estimated parameters

All the estimated elasticity and semi-elasticity coefficients (regression coefficients and intercepts) for each stressor–response relationship were stored in a database (Fig. 6 (step 3) and Supplementary Fig. 2b (step 3)).

### Step 4: A priori bias assessment and quality control

Data extracted from literature might favour certain response categories, for example stronger and 'significant' over weaker 'non-significant' responses. To assess this potential publication bias, we applied Egger's test, examining the shift in intercept based on the relationship between the inverse standard error and the ratio of parameter estimates to standard error (Fig. 6 (step 4)). Additionally, we analysed the distribution of z-values to identify systematic patterns indicative of selection bias[332]. These analyses revealed no clear bias, reinforcing the robustness of our estimates and minimizing the risk of overestimating stressor impacts (Supplementary Information (step 4)).

### Step 5: Prior formulation

A key element of our approach was the application of Bayesian model averaging (BMA) that allows for guiding models towards plausible stressor–response relationships based on prior information[55]. BMA requires the generation of several priors. In the Bayesian framework, posterior probabilities reflect updated priors of stressor–response relations given the likelihood of the data. To implement BMA, we generated four priors for each stressor–response relationship, classifying them into three sets: negative, neutral (unclear) or positive. This allowed us to incorporate directional expectations, such as the anticipated negative impact of salinity on freshwater biota. Details on prior selection are provided in Supplementary Information (step 5).

### Step 6: Meta-analysis

Applying the priors generated in step 5, we conducted a meta-analysis with the model parameters for each stressor–response relationship. We conducted a random-effects meta-analysis using BMA (Fig. 6 (step 6)) via R2JAGS. The Markov-chain-Monte-Carlo iterations were set to 30 chains with 3,000 iterations, thinned by 30. Model convergence was ensured by requiring: Rhat <1.01 and effective sample size >3,000. Bias adjustments were deemed unnecessary based on steps 4 and 7 (Supplementary Information (step 6)).

### Step 7: Posterior bias check

In addition to the a priori bias assessment (step 4) we conducted posterior bias checks following the meta-analysis (Fig. 6 (step 7)). We used funnel plots, which assessed posterior mean residuals as a function of the inverse of the standard error (1/s.e.). No diagonal patterns were observed, indicating no clear bias in stressor–response estimates (Supplementary Information (step 7)).

### Step 8: Posterior sensitivity check

We conducted a sensitivity analysis to evaluate the extent to which the priors influence the posterior estimates. To do this, we compared the posterior results presented in the main text with those from an alternative model using diffuse priors (priors without specific prior information) (Fig. 6 (step 8)). The analysis revealed that our prior assumptions about the stressor–response relationship tend to be more negative than the estimates derived from the data. Although some deviations from zero were observed, the estimated mode (the central tendency) remained stable (Supplementary Information (step 8)).

### Step 9: Visualization of stressor–response relationships

We generated several visualizations to enhance interpretability (Fig. 6 (step 9)). Posterior density plots (Fig. 2) illustrate plausible values for log-linear and logit-linear models, highlighting maximum-a-posteriori (MAP) estimates and 90% high-density intervals. HOPs (Fig. 3) visualize selected stressor–response combinations, modelling expected associations across stressor gradients while keeping other variables constant. Each HOP shows regression lines derived from posterior distributions, emphasizing variability. PDPs (Fig. 4) illustrate the marginal effects of two key stressors, aiding management-focused predictions (Supplementary Information (step 9)).

### Software and statistical packages

All analyses were performed in R. GLMMs were fitted with the glmmTMB package[333] (v.1.1.8) that handles random-effect structures in ecological data. The GAMLSS package[334] (v.5.4-20) was used for distributional analyses, while Bayesian modelling was conducted using JAGS (v.4.3.1) via the R2Jags package[335]. Data visualization was completed with ggplot2[336] (v.3.4.4), cowplot[337] (v.1.1.3) and bezier[338] (v.1.1.2).

### Reporting summary

Further information on research design is available in the Nature Portfolio Reporting Summary linked to this article.

## Data availability

Supplementary Data 1 provides details on all the articles and other data sources used in the meta-analysis and on the derived priors. Supplementary Data 2 lists the posterior estimations of all combinations of stressors and organism groups. Both files are available via GitHub at https://github.com/snwikaij/Data and Zenodo at https://doi.org/10.5281/zenodo.16947786 (ref. 339).

## Code availability

To support reproducibility, we developed an open-access R package, EcoPostView, enabling users to replicate, extend or visualize our analyses. This package allows custom data integration, alternative priors and posterior density visualization. The code and scripts used in this meta-analysis are available via GitHub at https://github.com/snwikaij/Data and the functions will be provided under the EcoPostView package via GitHub at https://github.com/snwikaij/EcoPostView.

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

## Acknowledgements

This paper results from the Collaborative Research Centre 1439 RESIST (multilevel response to stressor increase and decrease in stream ecosystems; www.sfb-resist.de) funded by the Deutsche Forschungsgemeinschaft (DFG, German Research Foundation; CRC 1439/1 and 1439/2, project number 426547801, phases I and II, received by B.S. and D.H.).

## Author contributions

W.K., C.S. and D.H. conceptualized the study. W.K. and C.S. curated the data. W.K., M.M., A.R.S., S.P., V.S.B., R.B., M.B., L.D., J.E., L. Farias, C.K.F., L. Feldhaus, S.M.G., K.H., S.E.O., S.L.M.O., I.M.P., C. Schlautmann, J.S., C.S.W., N.E.W., F.W. and C. Schürings screened the literature. W.K., M.M., A.R.S., S.P. and C. Schürings extracted the data from the literature. W.K. coded the models and performed the modelling. W.K., C. Schürings and D.H. wrote the original draft. All authors contributed to the refinement of the paper.

## Funding

## Competing interests

The authors declare no competing interests.

## Additional information

**Correspondence and requests for materials** should be addressed to Willem Kaijser, Christian Schürings or Daniel Hering.

**Publisher's note** Springer Nature remains neutral with regard
to jurisdictional claims in published maps and institutional
affiliations.

**Willem Kaijser** [1] ✉, **Michelle Musiol**[1,2,3], **Andrea R. Schneider**[1], **Sebastian Prati** [1,2], **Verena S. Brauer** [4], **Rike Bayer**[1],
**Sebastian Birk**[1,2], **Mario Brauns** [5], **Louisa Dunne**[1], **Julian Enss** [1,2], **Luan Farias** [1,2], **Christian K. Feld**[1,2], **Lena Feldhaus**[1],
**Svenja M. Gillmann** [1,2], **Kamil Hupało** [1,2,6], **Stephen E. Osakpolor**[7], **Sarah L. M. Olberg**[1], **Iris Madge Pimentel**[6],
**Ralf B. Schäfer** [3,8], **Christian Schlautmann**[1,2], **Jessica Schwelm**[1,2,9], **Bernd Sures** [1,2,3,9], **Cornelia S. Wagner**[1],
**Nicole E. Wells**[1], **Franziska Wenskus**[1], **Christian Schürings** [1,10] ✉ **& Daniel Hering** [1,2,10] ✉

[1]Aquatic Ecology, University of Duisburg-Essen, Essen, Germany. [2]Centre for Water and Environmental Research (ZWU), Essen, Germany. [3]Research
Center One Health Ruhr, University Alliance Ruhr, University of Duisburg-Essen, Essen, Germany. [4]Environmental Microbiology and Biotechnology,
University of Duisburg-Essen, Essen, Germany. [5]Department of River Ecology, Helmholtz Centre for Environmental Research—UFZ, Magdeburg, Germany.
[6]Aquatic Ecosystem Research, University of Duisburg-Essen, Essen, Germany. [7]Institute for Environmental Sciences, RPTU University of Kaiserslautern-
Landau, Landau, Germany. [8]Ecotoxicology, University of Duisburg-Essen, Essen, Germany. [9]Water Research Group, Unit for Environmental Sciences
and Management, North-West University, Potchefstroom, South Africa. [10]These authors jointly supervised this work: Christian Schürings, Daniel Hering.
✉e-mail: willem.kaijser@uni-due.de; christian.schuerings@uni-due.de; daniel.hering@uni-due.de

# Reporting Summary

## Statistics

For all statistical analyses, confirm that the following items are present in the figure legend, table legend, main text, or Methods section.

| n/a | Confirmed | |
|---|---|---|
| ☐ | ☒ | The exact sample size (*n*) for each experimental group/condition, given as a discrete number and unit of measurement |
| ☒ | ☐ | A statement on whether measurements were taken from distinct samples or whether the same sample was measured repeatedly |
| ☒ | ☐ | The statistical test(s) used AND whether they are one- or two-sided *Only common tests should be described solely by name; describe more complex techniques in the Methods section.* |
| ☒ | ☐ | A description of all covariates tested |
| ☐ | ☒ | A description of any assumptions or corrections, such as tests of normality and adjustment for multiple comparisons |
| ☐ | ☒ | A full description of the statistical parameters including central tendency (e.g. means) or other basic estimates (e.g. regression coefficient) AND variation (e.g. standard deviation) or associated estimates of uncertainty (e.g. confidence intervals) |
| ☒ | ☐ | For null hypothesis testing, the test statistic (e.g. *F*, *t*, *r*) with confidence intervals, effect sizes, degrees of freedom and *P* value noted *Give P values as exact values whenever suitable.* |
| ☐ | ☒ | For Bayesian analysis, information on the choice of priors and Markov chain Monte Carlo settings |
| ☐ | ☒ | For hierarchical and complex designs, identification of the appropriate level for tests and full reporting of outcomes |
| ☐ | ☒ | Estimates of effect sizes (e.g. Cohen's *d*, Pearson's *r*), indicating how they were calculated |

*Our web collection on statistics for biologists contains articles on many of the points above.*

## Software and code

Policy information about availability of computer code

| Data collection | R version 4.3.2, Rstudio, JAGS 4.3.1, WebOfScience |
|---|---|
| Data analysis | All data and code available from https://github.com/snwikaij/Data and https://github.com/snwikaij/EcoPostView |

For manuscripts utilizing custom algorithms or software that are central to the research but not yet described in published literature, software must be made available to editors and reviewers. We strongly encourage code deposition in a community repository (e.g. GitHub). See the Nature Portfolio guidelines for submitting code & software for further information.

## Data

Policy information about availability of data

All manuscripts must include a data availability statement. This statement should provide the following information, where applicable:
- Accession codes, unique identifiers, or web links for publicly available datasets
- A description of any restrictions on data availability
- For clinical datasets or third party data, please ensure that the statement adheres to our policy

All data is available from https://github.com/snwikaij/Data

# Research involving human participants, their data, or biological material

Policy information about studies with human participants or human data. See also policy information about sex, gender (identity/presentation), and sexual orientation and race, ethnicity and racism.

| | |
|---|---|
| Reporting on sex and gender | Not applicable |
| Reporting on race, ethnicity, or other socially relevant groupings | Not applicable |
| Population characteristics | Not applicable |
| Recruitment | Not applicable |
| Ethics oversight | Not applicable |

Note that full information on the approval of the study protocol must also be provided in the manuscript.

# Field-specific reporting

Please select the one below that is the best fit for your research. If you are not sure, read the appropriate sections before making your selection.

☐ Life sciences ☐ Behavioural & social sciences ☒ Ecological, evolutionary & environmental sciences

For a reference copy of the document with all sections, see nature.com/documents/nr-reporting-summary-flat.pdf

# Life sciences study design

All studies must disclose on these points even when the disclosure is negative.

| | |
|---|---|
| Sample size | *Describe how sample size was determined, detailing any statistical methods used to predetermine sample size OR if no sample-size calculation was performed, describe how sample sizes were chosen and provide a rationale for why these sample sizes are sufficient.* |
| Data exclusions | *Describe any data exclusions. If no data were excluded from the analyses, state so OR if data were excluded, describe the exclusions and the rationale behind them, indicating whether exclusion criteria were pre-established.* |
| Replication | *Describe the measures taken to verify the reproducibility of the experimental findings. If all attempts at replication were successful, confirm this OR if there are any findings that were not replicated or cannot be reproduced, note this and describe why.* |
| Randomization | *Describe how samples/organisms/participants were allocated into experimental groups. If allocation was not random, describe how covariates were controlled OR if this is not relevant to your study, explain why.* |
| Blinding | *Describe whether the investigators were blinded to group allocation during data collection and/or analysis. If blinding was not possible, describe why OR explain why blinding was not relevant to your study.* |

# Behavioural & social sciences study design

All studies must disclose on these points even when the disclosure is negative.

| | |
|---|---|
| Study description | *Briefly describe the study type including whether data are quantitative, qualitative, or mixed-methods (e.g. qualitative cross-sectional, quantitative experimental, mixed-methods case study).* |
| Research sample | *State the research sample (e.g. Harvard university undergraduates, villagers in rural India) and provide relevant demographic information (e.g. age, sex) and indicate whether the sample is representative. Provide a rationale for the study sample chosen. For studies involving existing datasets, please describe the dataset and source.* |
| Sampling strategy | *Describe the sampling procedure (e.g. random, snowball, stratified, convenience). Describe the statistical methods that were used to predetermine sample size OR if no sample-size calculation was performed, describe how sample sizes were chosen and provide a rationale for why these sample sizes are sufficient. For qualitative data, please indicate whether data saturation was considered, and what criteria were used to decide that no further sampling was needed.* |
| Data collection | *Provide details about the data collection procedure, including the instruments or devices used to record the data (e.g. pen and paper, computer, eye tracker, video or audio equipment) whether anyone was present besides the participant(s) and the researcher, and whether the researcher was blind to experimental condition and/or the study hypothesis during data collection.* |
| Timing | *Indicate the start and stop dates of data collection. If there is a gap between collection periods, state the dates for each sample cohort.* |

| Data exclusions | *If no data were excluded from the analyses, state so OR if data were excluded, provide the exact number of exclusions and the rationale behind them, indicating whether exclusion criteria were pre-established.* |
|---|---|
| Non-participation | *State how many participants dropped out/declined participation and the reason(s) given OR provide response rate OR state that no participants dropped out/declined participation.* |
| Randomization | *If participants were not allocated into experimental groups, state so OR describe how participants were allocated to groups, and if allocation was not random, describe how covariates were controlled.* |

# Ecological, evolutionary & environmental sciences study design

All studies must disclose on these points even when the disclosure is negative.

| Study description | We conduct a Bayesian Random Effects Meta-Analysis with Bayesian Model Averaging to quantify global stressor-response relationships across five riverine organism groups. Prior specifications are detailed in the appendix, with all priors provided in the Supplementary Information. |
|---|---|
| Research sample | Our study synthesizes 1,334 stressor-response relationships derived from 277 studies spanning 87 countries. These studies, identified through a systematic Web of Science search, encompass data on five key riverine organism groups - bacteria/archaea, algae, macrophytes, invertebrates, and fish—across seven prevalent stressors. |
| Sampling strategy | We included all studies from a systematic Web of Science search that met our inclusion criteria, yielding 1,334 stressor-response relationships from 277 studies. This comprehensive approach ensures broad taxonomic and geographic coverage, providing sufficient data for robust Bayesian meta-analysis. |
| Data collection | WK and CS collected data by systematically downloading studies from Web of Science, using various reference management tools, including Zotero. The literature was screened by WK, MM, ARS, SP, VSB, RK, MB, LD, JE, LF, CKF, LF, SMG, KH, SEO, SLMO, IMP, CS, JS, CSW, NEW, FW, and CS. Data extraction was carried out by WK, MM, ARS, SP, and CS according to predefined inclusion criteria. |
| Timing and spatial scale | Timing of data collection for this meta-analysis was not constrained, as it synthesized existing studies rather than conducting new sampling. |
| Data exclusions | Articles that did not fit the scope of the research questions or contained intractable information were excluded. No additional exclusions were made beyond these criteria. |
| Reproducibility | The results can be reproduced by running the code available from https://github.com/snwikaij/Data |
| Randomization | Randomization was not applicable to this study, as it is based on published literature available in Web of Science. The selection of studies was determined by predefined inclusion criteria rather than random allocation. |
| Blinding | Blinding was not applicable to this study, as it is a meta-analysis based on previously published literature. |

Did the study involve field work? ☐ Yes ☒ No

## Field work, collection and transport

| Field conditions | *Describe the study conditions for field work, providing relevant parameters (e.g. temperature, rainfall).* |
|---|---|
| Location | *State the location of the sampling or experiment, providing relevant parameters (e.g. latitude and longitude, elevation, water depth).* |
| Access & import/export | *Describe the efforts you have made to access habitats and to collect and import/export your samples in a responsible manner and in compliance with local, national and international laws, noting any permits that were obtained (give the name of the issuing authority, the date of issue, and any identifying information).* |
| Disturbance | *Describe any disturbance caused by the study and how it was minimized.* |

# Reporting for specific materials, systems and methods

We require information from authors about some types of materials, experimental systems and methods used in many studies. Here, indicate whether each material, system or method listed is relevant to your study. If you are not sure if a list item applies to your research, read the appropriate section before selecting a response.

## Materials & experimental systems

| n/a | Involved in the study |
|---|---|
| ☒ | Antibodies |
| ☒ | Eukaryotic cell lines |
| ☒ | Palaeontology and archaeology |
| ☒ | Animals and other organisms |
| ☒ | Clinical data |
| ☒ | Dual use research of concern |
| ☒ | Plants |

## Methods

| n/a | Involved in the study |
|---|---|
| ☒ | ChIP-seq |
| ☒ | Flow cytometry |
| ☒ | MRI-based neuroimaging |

# Antibodies

| Antibodies used | Describe all antibodies used in the study; as applicable, provide supplier name, catalog number, clone name, and lot number. |
|---|---|
| Validation | Describe the validation of each primary antibody for the species and application, noting any validation statements on the manufacturer's website, relevant citations, antibody profiles in online databases, or data provided in the manuscript. |

# Eukaryotic cell lines

Policy information about cell lines and Sex and Gender in Research

| Cell line source(s) | State the source of each cell line used and the sex of all primary cell lines and cells derived from human participants or vertebrate models. |
|---|---|
| Authentication | Describe the authentication procedures for each cell line used OR declare that none of the cell lines used were authenticated. |
| Mycoplasma contamination | Confirm that all cell lines tested negative for mycoplasma contamination OR describe the results of the testing for mycoplasma contamination OR declare that the cell lines were not tested for mycoplasma contamination. |
| Commonly misidentified lines (See ICLAC register) | Name any commonly misidentified cell lines used in the study and provide a rationale for their use. |

# Palaeontology and Archaeology

| Specimen provenance | Provide provenance information for specimens and describe permits that were obtained for the work (including the name of the issuing authority, the date of issue, and any identifying information). Permits should encompass collection and, where applicable, export. |
|---|---|
| Specimen deposition | Indicate where the specimens have been deposited to permit free access by other researchers. |
| Dating methods | If new dates are provided, describe how they were obtained (e.g. collection, storage, sample pretreatment and measurement), where they were obtained (i.e. lab name), the calibration program and the protocol for quality assurance OR state that no new dates are provided. |

☐ Tick this box to confirm that the raw and calibrated dates are available in the paper or in Supplementary Information.

| Ethics oversight | Identify the organization(s) that approved or provided guidance on the study protocol, OR state that no ethical approval or guidance was required and explain why not. |
|---|---|

Note that full information on the approval of the study protocol must also be provided in the manuscript.

# Animals and other research organisms

Policy information about studies involving animals; ARRIVE guidelines recommended for reporting animal research, and Sex and Gender in Research

| Laboratory animals | For laboratory animals, report species, strain and age OR state that the study did not involve laboratory animals. |
|---|---|
| Wild animals | Provide details on animals observed in or captured in the field; report species and age where possible. Describe how animals were caught and transported and what happened to captive animals after the study (if killed, explain why and describe method; if released, say where and when) OR state that the study did not involve wild animals. |
| Reporting on sex | Indicate if findings apply to only one sex; describe whether sex was considered in study design, methods used for assigning sex. Provide data disaggregated for sex where this information has been collected in the source data as appropriate; provide overall |

*numbers in this Reporting Summary. Please state if this information has not been collected. Report sex-based analyses where performed, justify reasons for lack of sex-based analysis.*

Field-collected samples | *For laboratory work with field-collected samples, describe all relevant parameters such as housing, maintenance, temperature, photoperiod and end-of-experiment protocol OR state that the study did not involve samples collected from the field.*

Ethics oversight | *Identify the organization(s) that approved or provided guidance on the study protocol, OR state that no ethical approval or guidance was required and explain why not.*

Note that full information on the approval of the study protocol must also be provided in the manuscript.

# Clinical data

Policy information about clinical studies

All manuscripts should comply with the ICMJE guidelines for publication of clinical research and a completed CONSORT checklist must be included with all submissions.

Clinical trial registration | *Provide the trial registration number from ClinicalTrials.gov or an equivalent agency.*

Study protocol | *Note where the full trial protocol can be accessed OR if not available, explain why.*

Data collection | *Describe the settings and locales of data collection, noting the time periods of recruitment and data collection.*

Outcomes | *Describe how you pre-defined primary and secondary outcome measures and how you assessed these measures.*

# Dual use research of concern

Policy information about dual use research of concern

## Hazards

Could the accidental, deliberate or reckless misuse of agents or technologies generated in the work, or the application of information presented in the manuscript, pose a threat to:

No | Yes
☐ ☐ Public health
☐ ☐ National security
☐ ☐ Crops and/or livestock
☐ ☐ Ecosystems
☐ ☐ Any other significant area

## Experiments of concern

Does the work involve any of these experiments of concern:

No | Yes
☒ ☐ Demonstrate how to render a vaccine ineffective
☐ ☐ Confer resistance to therapeutically useful antibiotics or antiviral agents
☐ ☐ Enhance the virulence of a pathogen or render a nonpathogen virulent
☐ ☐ Increase transmissibility of a pathogen
☐ ☐ Alter the host range of a pathogen
☐ ☐ Enable evasion of diagnostic/detection modalities
☐ ☐ Enable the weaponization of a biological agent or toxin
☐ ☐ Any other potentially harmful combination of experiments and agents

## Plants

| | |
|---|---|
| Seed stocks | Not applicable |
| Novel plant genotypes | Not applicable |
| Authentication | Not applicable |

## ChIP-seq

### Data deposition

☐ Confirm that both raw and final processed data have been deposited in a public database such as GEO.

☐ Confirm that you have deposited or provided access to graph files (e.g. BED files) for the called peaks.

| | |
|---|---|
| Data access links<br>*May remain private before publication.* | *For "Initial submission" or "Revised version" documents, provide reviewer access links. For your "Final submission" document, provide a link to the deposited data.* |
| Files in database submission | *Provide a list of all files available in the database submission.* |
| Genome browser session<br>(e.g. UCSC) | *Provide a link to an anonymized genome browser session for "Initial submission" and "Revised version" documents only, to enable peer review. Write "no longer applicable" for "Final submission" documents.* |

### Methodology

| | |
|---|---|
| Replicates | *Describe the experimental replicates, specifying number, type and replicate agreement.* |
| Sequencing depth | *Describe the sequencing depth for each experiment, providing the total number of reads, uniquely mapped reads, length of reads and whether they were paired- or single-end.* |
| Antibodies | *Describe the antibodies used for the ChIP-seq experiments; as applicable, provide supplier name, catalog number, clone name, and lot number.* |
| Peak calling parameters | *Specify the command line program and parameters used for read mapping and peak calling, including the ChIP, control and index files used.* |
| Data quality | *Describe the methods used to ensure data quality in full detail, including how many peaks are at FDR 5% and above 5-fold enrichment.* |
| Software | *Describe the software used to collect and analyze the ChIP-seq data. For custom code that has been deposited into a community repository, provide accession details.* |

## Flow Cytometry

### Plots

Confirm that:

☐ The axis labels state the marker and fluorochrome used (e.g. CD4-FITC).

☐ The axis scales are clearly visible. Include numbers along axes only for bottom left plot of group (a 'group' is an analysis of identical markers).

☐ All plots are contour plots with outliers or pseudocolor plots.

☐ A numerical value for number of cells or percentage (with statistics) is provided.

### Methodology

| | |
|---|---|
| Sample preparation | *Describe the sample preparation, detailing the biological source of the cells and any tissue processing steps used.* |
| Instrument | *Identify the instrument used for data collection, specifying make and model number.* |
| Software | *Describe the software used to collect and analyze the flow cytometry data. For custom code that has been deposited into a community repository, provide accession details.* |

| Cell population abundance | *Describe the abundance of the relevant cell populations within post-sort fractions, providing details on the purity of the samples and how it was determined.* |
|---|---|
| Gating strategy | *Describe the gating strategy used for all relevant experiments, specifying the preliminary FSC/SSC gates of the starting cell population, indicating where boundaries between "positive" and "negative" staining cell populations are defined.* |

☐ Tick this box to confirm that a figure exemplifying the gating strategy is provided in the Supplementary Information.

# Magnetic resonance imaging

## Experimental design

| Design type | *Indicate task or resting state; event-related or block design.* |
|---|---|
| Design specifications | *Specify the number of blocks, trials or experimental units per session and/or subject, and specify the length of each trial or block (if trials are blocked) and interval between trials.* |
| Behavioral performance measures | *State number and/or type of variables recorded (e.g. correct button press, response time) and what statistics were used to establish that the subjects were performing the task as expected (e.g. mean, range, and/or standard deviation across subjects).* |

## Acquisition

| Imaging type(s) | *Specify: functional, structural, diffusion, perfusion.* |
|---|---|
| Field strength | *Specify in Tesla* |
| Sequence & imaging parameters | *Specify the pulse sequence type (gradient echo, spin echo, etc.), imaging type (EPI, spiral, etc.), field of view, matrix size, slice thickness, orientation and TE/TR/flip angle.* |
| Area of acquisition | *State whether a whole brain scan was used OR define the area of acquisition, describing how the region was determined.* |

Diffusion MRI     ☐ Used     ☐ Not used

## Preprocessing

| Preprocessing software | *Provide detail on software version and revision number and on specific parameters (model/functions, brain extraction, segmentation, smoothing kernel size, etc.).* |
|---|---|
| Normalization | *If data were normalized/standardized, describe the approach(es): specify linear or non-linear and define image types used for transformation OR indicate that data were not normalized and explain rationale for lack of normalization.* |
| Normalization template | *Describe the template used for normalization/transformation, specifying subject space or group standardized space (e.g. original Talairach, MNI305, ICBM152) OR indicate that the data were not normalized.* |
| Noise and artifact removal | *Describe your procedure(s) for artifact and structured noise removal, specifying motion parameters, tissue signals and physiological signals (heart rate, respiration).* |
| Volume censoring | *Define your software and/or method and criteria for volume censoring, and state the extent of such censoring.* |

## Statistical modeling & inference

| Model type and settings | *Specify type (mass univariate, multivariate, RSA, predictive, etc.) and describe essential details of the model at the first and second levels (e.g. fixed, random or mixed effects; drift or auto-correlation).* |
|---|---|
| Effect(s) tested | *Define precise effect in terms of the task or stimulus conditions instead of psychological concepts and indicate whether ANOVA or factorial designs were used.* |

Specify type of analysis:     ☐ Whole brain     ☐ ROI-based     ☐ Both

Statistic type for inference     *Specify voxel-wise or cluster-wise and report all relevant parameters for cluster-wise methods.*

(See Eklund et al. 2016)

Correction     *Describe the type of correction and how it is obtained for multiple comparisons (e.g. FWE, FDR, permutation or Monte Carlo).*

## Models & analysis

| n/a | Involved in the study |
|-----|----------------------|
| ☐ | ☐ Functional and/or effective connectivity |
| ☐ | ☐ Graph analysis |
| ☐ | ☐ Multivariate modeling or predictive analysis |

**Functional and/or effective connectivity**

*Report the measures of dependence used and the model details (e.g. Pearson correlation, partial correlation, mutual information).*

**Graph analysis**

*Report the dependent variable and connectivity measure, specifying weighted graph or binarized graph, subject- or group-level, and the global and/or node summaries used (e.g. clustering coefficient, efficiency, etc.).*

**Multivariate modeling and predictive analysis**

*Specify independent variables, features extraction and dimension reduction, model, training and evaluation metrics.*

