## [Peer Review File · Nature Ecology & Evolution]

Global scale quantification of stressor responses in five riverine organism groups

Corresponding Author: Dr Daniel Hering

Version 0:

Decision Letter:

14th May 2025

Dear Dr Hering,

Your manuscript entitled "Global scale quantification of stressor responses in five riverine organism groups" has now been seen by 3 reviewers, whose comments are attached. The reviewers have raised a number of concerns which will need to be addressed before we can offer publication in Nature Ecology & Evolution. We will therefore need to see your responses to the criticisms raised and to some editorial concerns, along with a revised manuscript, before we can reach a final decision regarding publication.

We therefore invite you to revise your manuscript taking into account all reviewer and editor comments. Please highlight all changes in the manuscript text file in Microsoft Word format.

Please note that our methods sections are not word limited, and the main article (not including methods) can be up to 3500 words. We also can't have display items in the methods only, so they should be cited in the main text if possible.

* If you have not done so already please begin to revise your manuscript so that it conforms to our Article format instructions at <http://www.nature.com/natecolevol/info/final-submission>. Refer also to any guidelines provided in this letter.

* Extended Data Figures - please ensure that any supplementary figures and tables that are crucial to the manuscript's conclusions are converted into Extended Data figures and tables to increase visibility of these data. Extended Data figures and tables are online-only (present in the online PDF and full-text HTML versions of the paper), peer-reviewed display items that provide essential background to the article but are not included in the main article due to space constraints. A maximum of ten Extended Data display items (figures and tables) is permitted.

Link Redacted

Nature Ecology & Evolution is committed to improving transparency in authorship. As part of our efforts in this direction, we are now requesting that all authors identified as 'corresponding author' on published papers create and link their Open Researcher and Contributor Identifier (ORCID) with their account on the Manuscript Tracking System (MTS), prior to acceptance. ORCID helps the scientific community achieve unambiguous attribution of all scholarly contributions. You can create and link your ORCID from the home page of the MTS by clicking on 'Modify my Springer Nature account'. For more information please visit www.springernature.com/orcid.

[redacted]

Reviewer expertise:

Reviewer #1: ecology, biodiversity trends

Reviewer #2: freshwater ecology, global change

Reviewer #3: freshwater ecology

Reviewers' comments:

Reviewer #1 (Remarks to the Author):

Here the authors conduct an impressive analysis of the impacts of different freshwater stressors on several taxon groups. They integrate information from over 200 studies to generalize these relationships and identify which stressors matter the most for each group. Given the variety of threats to freshwater systems, this paper has broad implications and a potentially broad readership.

The Introduction section is very well written, but I found it hard to follow the Methods section and understand what sort of data were used, which made it hard to review. I have made various suggestions for improvements.

Major comments:

- Quite a few aspects of the methods need more description in the main text. I found I had to first read the SI to understand, but even then I remained lost about the modelling approach. A general suggestion is to explain the purpose of each method component before giving details on it. The methods often zoom in on technical details and omit some of the key high-level aspects that would help guide a reader.
- For instance, inclusion criteria: Were both field-based experiments and field observation studies included? Obviously, the latter can suffer from issues about causal inference of the stressor effects, so at the very least a difference between them should be tested. If observational studies were included, how were potentially confounders dealt with? If an experiment study was multifactorial, how were the other factors dealt with?
- We need a better description of the underlying dataset (a brief overview of time- and spatial-scale of each study, whether they were assess variation along spatial or temporal gradients, typical number of sampling sites / rivers per study etc..)
- Statistical analysis: Unclear how the data were grouped to analyse in the glmms. Usually in a synthesis, there is either a two-step approach (producing effect sizes per study and then synthesis of them in a meta-analysis) or a one-step approach (one large synthesis model). Here, we seem to do something in the middle and incorporate both? Rather unclear though.
- You mentioned (e.g., line 276) you estimate the change in the responses due to % change in each stressor. But does % increases really make sense for variables like temperature? What would a 50% increase in temperature mean – it means different things across different studies and gradients? I can see why you did this – since the stressors are in different units. But % is a measure of multiplicative change, I am not sure that makes sense for all variables. An alternative approach would have been to normalize the gradients between minimum and maximum values, or to scale in units of standard deviations. Either consider these alternatives or better justify your approach.
- The results focus on the results in binary terms – positive/negative – I think it would be more interesting to focus on the relative magnitudes of the stressors, since this is the key area of uncertainty (this comparison could be aided by the same axes scale in Fig. 2 or in a different figure). Also, as in most meta-analysis, it would be nice to report measures of heterogeneity -i.e., are the relationships more variable for some stressors than others? Currently, the focus is rather on the mean effect only, but there is a lot more information here potentially, relevant for interpretation.

Minor comments:

Line 34: potentially misleading – here you report 22,120 studies but later only 277 (line 93).

Line 46: I am not sure this blanket statement is fair and representative: freshwater insects have done better than many others e.g., Haase et al. 2023 Nature; van Klink et al. 2020 Science. Some nuance is needed here.

Line 77 – since this nature paper just came out (The global human impact on biodiversity | Nature), you'll also need to

distinguish yourself from that.

Line 79 – make clear whether you include experimental and/or observation data here

Line 79 - in fact, I'd say here include the specific set of inclusion criteria that you used to determine the relevance of a study. Also some sort of statement on how you made the results comparable across different studies (presumably they varied in the magnitude of their stressor gradients. I appreciate its an NEE paper, but a bit more on methods is needed, but the results can be interpreted.

Line 100 – since this is the largest pool, report the % for richness and abundance separately.

Methods

Line 257 – so you kept both field experiments and observational field studies. Any difference between these should be tested since each typically vary in internal vs external validity.

For observation field studies, was there any attempt to control for confounders when assessing the impact of any specific driver? Was the gradient a spatial gradient (e.g., sites in different temperatures) or a temporal gradient (e.g., temperatures vary among years). The latter is more likely to be a causal in most observational studies since spatial gradients of different variables typically covary with each other.

Line 265 – I expect some information here on data extraction – i.e.. what information did you extract from each study. Its unclear at this stage whether you were seeking essentially the raw data from the studies, or whether you wished to directly extract coefficients for the effects of different stressors from the studies, or summary variables.

Line 269 – as mentioned above, I am unclear what a dataset would look like. Is it e.g., species richness under different values of a stressor gradient? Did you include studies treating both stressor as a factor versus a continuous variable?

Spatial and temporal gradients?

Line 269 – not sure what 'dataset' refers to here. Is it per study? Or per taxon/response group? Or?

Line 273 – mention what the independent variables were. Did you build separate models for each stressor? Or use multiple regression? If a study included 'control' variables e.g. sampling effort, did you also include those?

Line 287 – Not sure what you mean by 'accumulation'. Synthesis? But that would be confusing since you already have a later section on meta-analysis.

Line 321 – since you used quite strong priors, this sensitivity analysis using uninformative priors is important. Here you are reporting the results? I don't follow what you mean. The results were the same or? Either way, mention in the main text.

Figures and Tables

I'd rather have Fig 1 as a panel plot – you show geographic coverage, but I'd include for b. taxonomic coverage and c. stressor coverage across the studies. (Table 1 could then be moved to SI – especially for NEE, I'd go for plots than tables for easier communication).

Figure 4 – the models assumed additive effects? In that case, I don't think it makes sense to show 2-D covariate plots.

Figure 5 - I don't think its so helpful to specify the number per model type (log-linear or logit) as per bottom row. Of more interest biologically, is the number of each response variable.

Figure 6 showing the analytical workflow is important but the diagrams are not informative enough. I suggest adding more text to explain the steps (and use the step numbers in the text more).

SI comments

Line 86 – I am not sure what you mean here by false-negative rates nor how it was assessed. This whole para needs more detail on rationale and methods.

Line 88 – who did the analysts receive feedback from? Each other?

Line 96 – I would separate inclusion criteria (what determines whether a study was included or not) and data extraction (what was extracted from included studies) into separate sections

Line 101 – what biodiversity metrics were of interest? Also mention here the inclusion criteria on study design and stressor types of interest.

Line 111-113 – do you mean to imply that %EPT-taxa and coverage% were treated as evenness metrics. How do you justify this?

Line 103 – since we haven't started talking about models at this point, I'd just frame this section as response variables of interest – the distinction between logit and log should come into the next section about model fitting.

Line 106 – the main text also mentioned abundance (line 100)

Line 116 – but how does such a categorization help assess bias? And in any case what sort of bias are we meaning? I think this sentence needs to be written either way.

Line 119 - is this fitting models to each dataset/study separately? Or are we already doing synthesis (line 81 in your main text indicates that the glmms and meta-analysis were two separate things, but now I am not sure)

Line 126 – obviously seasonal effects vary across the globe, but you are modelling it as a constant variation here as a random effect - whether your approach is valid here depends on whether these are separate models for each dataset/study or not? If these are already synthetic models, the seasonal effect could be allowed to vary among studies/regions (actually at this point, I realise I am unclear on how long these studies typically are – annual long-term or short-term daily measurement studies? Some description of this is needed, also how many sampling sites/rivers does each study tend to have?).

Lines 129-131 comes as a surprise since we seem to be modelling all variables at the same time. This was not clear before. So was another inclusion criteria that a study measured multiple stressors? Why isn't dataset or study mentioned as a random effect? As you might be gathering by now, I am rather confused by the models!

Line 153 – what does accumulation mean? Synthesis? Aggregation? Not sure what is happening in this step. Is it processing the model outputs?

Line 167 – what sort of bias? Publication bias?

Line 173 – so you have multiple values per study - I am still unclear what they are and what you are aggregating in the same

model here. Also - why would source (figure or table) be relevant?

Line 193 – according to what assumptions should this be normally distributed-explain the thinking more.

Line 201 – we are talking about priors before talking about what any Bayesian model would seek to do... Is this about the earlier models (if so it should be moved there, but it wasn't mentioned that the GLMMs would be fit within a Bayesian framework) or the next models for the meta-analysis?

Line 223 – write an opening section about what you seek to do with these priors? Are you trying to integrate existing knowledge? Given you have a lot of data, I would have suspected that the priors are less important here, so I am unclear why you take a more complex approach than many similar syntheses (that usually use uninformative priors). Again, some rationale sentences would be helpful.

Reviewer #1 (Remarks on code availability):

Not done, since paper needs re-review.

Reviewer #2 (Remarks to the Author):

In general, this is an interesting and quite unique contribution that seeks to identify broad relationships between putative stressors and different taxonomic groups of freshwater organisms at a global scale. I think the work will attract broad interest, but needs:

i) Some attention to the clarity of the writing in places (expanded below)

ii) Recognition that some important potential stressors have not been included – for example invasive non-native species and emerging contaminants

iii) Recognition that some pressures on freshwater organisms could arise from life stages in other systems – for example marine ecosystems in the case of migrant fishes or terrestrial systems in the case on insects with flying adult stages

iv) Some more clarity with respect to the nature of the relationships detected – for example whether they are linear, curvilinear etc

I think there are likely to be taxonomic challenges in understanding the data. While biodiversity is relatively straightforward to appraise in, say, fishes or macrophytes, this is not so easy in groups such as bacteria or archaea – where identification depends on features such as gene sequences or even gene expression

Even with relatively large data sets (as here) – I wondered also about intrinsic biases in the global data available? Examples are where invertebrates or fishes are likely to more readily sampled (and therefore more often studied) from lower-order river systems than from large rivers – where stressor combinations and effects could change.

I also wonder what detail may be masked at the level of taxonomic aggregation used here. The prime example would be the lumping, say, of bryophytes and angiosperms as 'macrophytes' – but these two have very different evolutionary origins and environmental requirements. Similarly, responses to pressures in different sub-groups of the other taxa (fishes, invertebrates...) are likely to vary strongly with adaptation in ways that cannot be captured with such aggregated analysis.

On points of detail:

Lines 32, 33 and 37: could you indicate which aspects of biodiversity you addressed? Richness? Alpha? Beta...?

Lines 39/40: The multiple adjective makes this slightly hard to read (critical, multivariable, stressor-response. And tailored management strategies for what purpose?

Line 43: can you genuinely extrapolate from your analysis to an understanding of resilience?

Line 48: this feels in need of some expression of temporal as well as spatial scales: global biodiversity decline is a process through time.

Line 47-55. I respect that these are examples of global change pressures – but I wonder at the global effect of invasive non-native species or the overlooked effects of emerging contaminants? These are poorly parameterised in many studies. Perhaps a point for discussion.

Lines 58 and 59: should these numbers be reflected by the names of the authors responsible?

Lines 66-68: what of biotic stressors – where species interactions are involved? There is some evidence that biodiversity decline can be mediated by interactions between global change processes and predator/prey or competitive relationships. (eg <https://www.journals.uchicago.edu/doi/10.1899/09-159.1>)

Line 73: there are some complications, for example where fishes are long-distance migrants that move between realms or where insects have aerial stages that interact with terrestrial ecosystems.

Line 78/79: there are some taxonomic challenges in understanding the focal units being assessed – for example between bacteria versus fish

Line 110: and the difficulty of understanding 'taxonomy' in micro-organisms?

Line 124-134: this paragraph could be stronger. First, highly saline marine ecosystems have rich algal communities – so the effects of salinity may not be so straightforward. Second, the nature of the relationship between nutrients and algal richness is not made clear: is it linear or curvilinear? Third, lines 132-134 illustrate the challenges of assessing diversity in different groups where methods are not standardised or assessments made in ways that don't account for sample size (eg through

rarefaction)

Line 133-134: the logic here could be clearer

Line 135-140: How are 'macrophytes' defined? It's likely that the patterns described might differ, say, between bryophytes and angiosperms

Line 141 and 142/143: these lines clearly overlap

Line 144-146: the logic could be clearer: why does this imply an indirect relationship? And in what sense?

Line 149-151: this contrast with invertebrates is interesting: does it have evolutionary origins? Fishes colonised freshwaters from marine environments while many invertebrates (especially insects) colonised freshwaters from terrestrial realms.

Line 176: threat

S J Ormerod, Cardiff April 10 2025

Reviewer #3 (Remarks to the Author):

General:

This synthesis presents some very interesting and relevant data that would be of interest to a broad readership and could have important implications for informing management directions of river systems. To achieve this, however, the descriptions and discussion of the results require some work to tell a cohesive story that is fully supported by the data. I encourage the authors to consider restructuring the discussion of their results around the key stressors of interest, rather than paragraphs summarising responses of each taxonomic group. Exploring some of the strongest stressor responses in more detail (as shown for the selected HOPs and PDPs, particularly for important stressor interactions such as nutrients x warming, sedimentation x flow velocity, for example) and highlighting key management implications that result from these analyses would considerably enhance the value of this paper. With some revision, there is certainly potential for an interesting and informative piece that provides important insights for conservation strategies and highlights the potential for future analyses utilising similar approaches, as per the concluding sentence of the abstract. I have provided specific comments below that I hope will be of use in improving the manuscript.

Specific comments:

Main text.

L55: Here and throughout, I personally find the use of the term "relations" a bit strange, to me this is more a term used with respect to people, e.g. family, and suggest "relationships" which, while almost the same word, I am more familiar with in scientific terms. Elsewhere I have suggested using "responses", instead of relations.

L61: Suggest altering "aimed at unravelling" to just "investigated".

L62-3: Possibly something briefly discussing the challenges of developing such a predictive framework would be useful.

L64-6: It is not clear to me why this is the case. Orr et al. 2024 showed there are almost 2400 studies investigating the individual and combined effects of multiple stressors, generally on specific organisms? So the meaning behind this paradox is unclear.

- Further comment: I think the confusion for me lies in the use of "specific organism groups", because I initially read this as taxonomic groups at e.g. species-to-family level. But given the focus of the meta-analysis on broad groups as described in L77-9, I think this is really what the authors are getting at with respect to the paradox - that the focus on specific taxonomic groups has obscured the interpretation of absolute effects on broader groups?

L66-8: What about biological stressors? Invasive species, pathogens, biotoxins, etc.

L68-71: "Each stressor operates through distinct mechanisms"; However, these mechanisms are not always known or well understood. Moreover, there are many different levels of these mechanisms - the example of fine sediment accumulation or channelisation altering habitat availability is a very broad mechanism which has many sub-mechanisms that will result in which species are favoured vs excluded. These are quite different to the specific cellular mechanisms of osmoregulation and slowed metabolism mentioned in the previous sentence. Possibly some brief discussion of this issue could be useful (i.e. that the mechanisms of operation/mode of action of stressors can operate at different levels, and how some stressors operate are more well understood than others at different levels - from the cellular to habitat scale).

L74: Suggest "broader" instead of "broad", Using "while" to begin the second half of the sentence suggests a comparison to be made with the first half.

L107-9: Can directionality of effects be added? Generally strong negative responses to stressors, except for Phosphorus where both groups responded positively, and warming where fish responded positively (and invertebrates were the only taxonomic group to respond negatively)?

L109: With slightly more description added to the first sentence, the additional detail might be needed specifying the contrast, i.e., “In contrast to the microorganism groups, microorganisms...”

L109-11: I'm not sure this conclusion holds: For taxonomic richness, the variability in responses to stressors is equal for bacteria, macrophytes and fish in terms of 2/7 negative or positive effects. Likewise for evenness, bacteria, algae, macrophytes and fish all have 3/7 positive or negative effects. So really shouldn't the conclusion be something like; consistency in response to stress is really only observed among invertebrates, which are negatively affected by all stressors except for phosphorus enrichment, thus highlighting their importance as biological indicators of stress/pollution/habitat degradation etc.?

I also note that some of this is subsequently picked up in, e.g., paragraph 135-140 on macrophyte responses, where the contrasting positive and negative effects to different stressors is discussed, as well as the paragraph beginning L147 on fish; “Fish exhibit a mix of positive and negative stressor-response relationships.”

L111-113: Again, a conclusion that doesn't appear to be fully supported by the results – macrophytes showed the same (negative) response to all taxonomic groups for salinity-increase (except bacteria), the same positive response to warming as all groups (except invertebrates), the same response to P-increase re. taxon richness, though evenness showed the opposite response (could this surprising result be affected by assigning a negative prior set to macrophytes for phosphorus – see further comment below), as was also the case for bacteria. These responses highlight the nuance that this conclusion isn't capturing and could be misleading. Suggest a more specific description where this conclusion holds, such as for oxygen-depletion and flow cessation which align more closely with the discussion of their unique adaptations as rooted, sessile autotrophs.

L116-7: As mentioned above – not just bacteria? Shouldn't this description be; “notably all groups other than invertebrates exhibited a positive relationship with temperature.” This seems a somewhat surprising result and warrants further discussion and exploration of why; one could argue according to this study that climate warming will positively affect freshwater organisms other than invertebrates...

L141: remove comma after “both,”

L145-6: This is a surprising result given previous studies, including meta-analyses with some of the same authors, showed that nutrient enrichment was a key driver in invertebrate responses/distributions (Birk et al. 2020).

L147: As noted previously, suggest using “responses” rather than “relations”.

Fig. 2: The different colours are useful for indicating positive vs negative estimates in a busy plot. Adding some small icons for each taxonomic group and stressor could also be a useful visual tool to aid interpretability.

Fig. 3: panel a) x-axis needs to be corrected to Nutrient-N (mg/L).

L165-77: Why were these HOPs chosen for deeper discussion? It would be interesting to see the response of invertebrates to temperature, given their contrasting response to all other groups for this stressor. Likewise, a plot comparing the strong positive response of fish to warming would be interesting. Ultimately, a separate panel figure for each key stressor, such as one each for nitrogen, phosphorus, warming and fine sediment, with all taxonomic groups displayed could be quite useful. Perhaps one could be chosen for the main text and the others included in supporting information.

L178-89: Similarly, to the above comment, why these two plots? Fine sediment and flow velocity would be an interesting combination, given their known mechanistic interactive potential. Or temperature with other stressors (e.g. nutrients) given efforts to mitigate climate warming, despite positive effects being observed for most groups...potentially fine sediment and warming would be an interesting PDP for discussion/management implications. I'm not sure that fine sediment and oxygen is so useful, as the discussion in L188-9 “mitigating sediment inputs alone may be insufficient unless accompanied by efforts to maintain adequate oxygen levels” does not have a clear management application, given maintaining oxygen levels is difficult in practice and is more likely achieved by reducing nitrogen/phosphorus loading which contributes to exacerbated diurnal DO fluctuations and, potentially, overnight hypoxic conditions.

The importance of DO for many invertebrate taxa also doesn't seem to be reflected in the PDP, where one might expect the bottom-left corner to be more pink showing reduced richness under low oxygen (i.e. more like the colour gradient of the nutrient-salinity PDP), perhaps this is because the gradient stops at 5 mg/L and therefore doesn't reflect fully hypoxic conditions – could the gradient be extended? Currently, the concluding sentence could actually be interpreted in the opposite way, given that invertebrate richness is still relatively high (35-40) at the lowest oxygen level shown. I.e. “Our results suggest that mitigating fine sediment inputs could strongly improve invertebrate richness, even under low oxygen conditions, highlighting the pervasive effects of this stressor and the importance of management implications to control its entry into freshwaters” might be a more valid and important conclusion to draw from this PDP.

L180-1: Describing the nutrient-salinity PDP in the context of increased salinity offsetting benefits of nitrogen enrichment on algal evenness also seems like an interesting choice description and plot to present among the many options, given most management strategies in freshwater systems seek to mitigate nitrogen loading due to effects described in the previous comment (increased algal growth leading to greater diurnal DO cycles and overnight hypoxia).

Concerning the description, using the term “counteracts” rather than “offsets” might be better in the context of counteracting benefits of nutrients, whereas offsetting is likely better used when affecting negative effects.

Beyond this, however, I would again suggest the gradient of this PDP could be interpreted differently. The slope of this

gradient could suggest that nutrient enrichment offsets the negative effects of raised salinity, given the upper-right corner at very high salinity levels is not as pink as might otherwise be expected (evenness is still around the midpoint of the gradient at ~0.3)

L181-3: This sentence might fit better at the end of this paragraph, although I'm not sure it's needed.

A general comment on this results section: Possibly choosing either evenness or richness for the HOPs and PDPs would aid interpretability, as it allows the reader to consistently think of the same response when looking across plots. Plots for the other response could be included in Supp infos.

I also suggest changing or removing the subheading "Macro- and microorganisms exhibit divergent responses to stressors" as this isn't really that informative, and I'm also not sure it's the most interesting result to highlight.

I invite the authors to consider a revised results section that is structured around the stressors, rather than the organism groups. This might streamline the descriptions and reduce the need for some of the contradicting sentences with respect to which taxonomic groups show the most divergent responses to stressors. Of course, different taxonomic groups might have different responses to stress, given their different functional roles at various trophic levels. What is more interesting is how the different stressors manifest effects, at what strength and in which direction across different trophic levels in an ecosystem and which taxonomic groups are more strongly affected (positively or negatively) and why. Thus, a stressor-structured results section would, in my opinion, be more interesting and relevant for management implications that must consider the whole ecosystem, not just separate taxonomic groups. This is the discussion I would expect to read in a "Global scale quantification of stressor responses (in five riverine organism groups)" – the text in parentheses could also be changed to "in riverine ecosystems", the important part is the quantification of stressor responses.

L202 (Section; Research and management implications):

I think this section would flow better starting with the discussion of the stressor-response relationships relevant to management as discussed in paragraph(s) beginning L225. The paragraph(s) beginning L203 would then fit better toward the end of this section, as they are discussing implications and future directions for the research field, and begin with "Beyond estimating the stressor-response relationships", therefore it would make sense to have already discussed these. If this order is used, the subheading could be re-arranged to "Management and research implications" to reflect the section structure.

Supp. information.

L297 – Why did macrophytes get assigned a negative prior set for nutrients-N and P? Would they not respond positively to nutrient addition?

L338-9: It seems surprising that the main model for bacteria and sediment enrichment would have a much more negative logged odds ratio compared to the uninformative model, given that neutral priors were used for bacteria. What is causing this result?

L340-1: The trend seems to be most evident in fish which are not listed. Whereas some of the listed groups show neutral or even positive differences between M1 and M0, e.g. for oxygen-depletion, sediment-enrichment for algae and macrophytes and N/P-increase.

L352-4: Repeated sentences from an edited version – remove one version.

Reviewer #3 (Remarks on code availability):

README provided, reproducibility good.

*****END*****

Version 1:

Decision Letter:

23rd July 2025

Dear Dr. Hering,

Thank you for submitting your revised manuscript "Global scale quantification of stressor responses in five riverine organism groups" (NATECOLEVOL-25020579A). It has now been seen again by the original reviewers and their comments are below. The reviewers find that the paper has improved in revision, and therefore we'll be happy in principle to publish it in Nature Ecology & Evolution, pending minor revisions to satisfy the reviewers' final requests and to comply with our editorial and formatting guidelines.

If you have not done so already, please ensure that you also email us a completed copy of the Reporting summary :

Reporting summary: <https://www.nature.com/documents/nr-reporting-summary.pdf>

[redacted]

Reviewer #1 (Remarks to the Author):

Many thanks for carefully considering my comments – the methods are now much clearer to me. As before, I feel this is a well-written, well-conducted and important piece of work, which I expect to have significant impact.

I do have a couple of outstanding comments:

Data sources

I appreciate the additions to the SI about the data included. You've now well-described the aggregate dataset. But I am missing something for the main text. I expect to see a sentence something like this:

"Each study sampled a riverine community at least 5 times (mean = X), across at least X sites (mean=X) in X years (mean =X), focusing on at least one stressor (mean = X stressors)".

This could go somewhere around line 88.

Temporal vs spatial

Apologies if my last comment was not clear enough, but its still not clear whether your datasets study temporal or spatial gradients in stressors. E.g., is the x-axis for the stressor effect within a dataset (step 1) representing different sites/rivers (each with different stressor values) within the same year, or different years (with different stressor values) at the same sites. Its well-known that different strengths of relationships can be found since different ecological processes affect each (with lags especially affecting temporal effects, and confounding especially affecting spatial effects - hence the whole space-for-time substitution debate). A paper unpacks it here: Birds and Climate in Space and Time: Separating Spatial and Temporal Effects of Climate Change on Wildlife – Methods Blog

I appreciate your paper is not delving into such detail and I am not suggesting you follow exactly the methods of this paper. But it still needs to be clear whether the studies focus on spatial or temporal stressor variation. This is linked to my comment above about the summary of the data.

Based on your current inclusion criteria (at least 5 obs – without specifying the units of those obs across space or time), there is the impression that you have both temporal and spatial studies; though I am guessing probably mostly the latter. Spatial gradients are typically easier to study (within a typical short-term project), but for many questions e.g., making predictions about future change, we want to make inferences about temporal changes (again, linking with the space-for-time issue). If you do have a mix of studies, then this could explain some variation in your results, and so ideally would have been tested or controlled for (by labelling each study as 'spatial' or 'temporal'). If this is not possible, this is a limitation that could be briefly mentioned in your Discussion, and unpacked in future work.

Minor

Line 120-121 – is bold needed here?

Line 88 – 'uncertainty' instead of 'clarity'

Reviewer #2 (Remarks to the Author):

In general, an improved version which has incorporated caveats over points made on the initial submission. I've concentrated mostly on editorial issues that could be addressed to help readability.

Line 37: 'taxa richness' is not grammatically correct despite being in wide use in the literature - particularly from N America. Taxa are plural, taxon is singular. You mean taxon richness.

Line 37. Relation with, not relation of

Line 26: shouldn't this say that reduced biodiversity (...) reflected elevated salinity, oxygen depletion, and fine sediment

accumulation? As written, the implication is that biodiversity affected physicochemical conditions rather than the reverse

Lines 45-47 Here and in other places, the writing could be simpler. "Over recent decades, freshwater ecosystems - particularly rivers - have experienced rapid biodiversity decline caused by multiple, interacting stressors at local, catchment and global scales"

Line 50: agricultural intensification?

Line 68-71: could be simplified

Line 120-122: something wrong here

Line 126: I'd use the past tense to maintain consistency with previous sentences

Line 129-130: Perhaps: 'However, our meta-analysis necessarily obscures divergent patterns of macrophytic taxa, such as bryophytes and vascular plants, which vary substantially in evolutionary origins, traits and environmental requirements'.

Line 139: are or were? Past tense? You have some tendency to switch between tenses.

Line 147: I think this sentence structure mixes dependent and independent variables - here and elsewhere

Line 187-200: past tense for results?

Line 225: were?

S J Ormerod, Cardiff 13/07/25

Reviewer #3 (Remarks to the Author):

I have re-read this manuscript and reply letter and I am satisfied that the authors have appropriately revised the manuscript, and sufficiently addressed all of my comments and suggested revisions.

Reviewer #1

No	Comment	Response
1	Here the authors conduct an impressive analysis of the impacts of different freshwater stressors on several taxon groups. They integrate information from over 200 studies to generalize these relationships and identify which stressors matter the most for each group. Given the variety of threats to freshwater systems, this paper has broad implications and a potentially broad readership. The Introduction section is very well written, but I found it hard to follow the Methods section and understand what sort of data were used, which made it hard to review. I have made various suggestions for improvements.	Thank you! We have modified the Methods section in many places following your (and the other reviewers') suggestions.
2	Quite a few aspects of the methods need more description in the main text. I found I had to first read the SI to understand, but even then I remained lost about the modelling approach. A general suggestion is to explain the purpose of each method component before giving details on it. The methods often zoom in on technical details and omit some of the key high-level aspects that would help guide a reader.	Thank you for this suggestion. We agree that the methods are complex and have rarely been used in ecological research, but we firmly believe they are most suited for highly variable datasets as used in this study. We further agree that the methodological descriptions require more context. Following your suggestion, we have now added a sentence at the beginning of each paragraph of the "Methods" chapter, highlighting the purpose of the respective methods. For instance, for the paragraph on "Model fitting and parameter extraction" we now start with the sentence: "The purpose of this step was to derive the model parameters and standard errors of the Generalized Linear Mixed Models (GLMM) between each stressor and biotic response (...)"
3	For instance, inclusion criteria: Were both field-based experiments and field observation studies included? Obviously, the latter can suffer from issues about causal	Thank you for this important comment. We would like to clarify that our study included only observational field studies; no experimental studies were included. This was already obvious from the search terms used for the queries in WebOfScience (SI, Step 1.) This choice was deliberate, as our primary interest lies in understanding how stressors affect biota under real

	inference of the stressor effects, so at the very least a difference between them should be tested. If observational studies were included, how were potentially confounders dealt with? If an experiment study was multifactorial, how were the other factors dealt with?	world conditions, i.e. in the field, as reflected by field surveys and observational datasets. This decision and its rationale are now clearly stated in paragraph 4 of the main section. (“We did not include experimental studies, as our prime interest lies in the relationship between stressors and biota under real-world conditions.”) As we did not include data from experimental studies, there was no need to deal with “other factors” used in multifactorial studies. We fully acknowledge that observational studies present challenges for causal inference. However, our objective is not to establish causality per se, but rather to examine empirical relationships between stressor intensity and biodiversity metrics such as species richness and evenness. We have added a sentence in paragraph 5 of the main chapter to better clarify our intentions: “The prime objective was to establish empirical relationships between stressor intensity and biodiversity metrics over a wide range of conditions, independently of possible causes.” To reflect this, we intentionally avoid causal terminology. For example, we use the phrase “stressor–response relationship” rather than “stressor effect” - a subtle but deliberate distinction, underscoring that our findings should not be interpreted as causal effects in the strict sense. To further elaborate, establishing causality from observational data is inherently difficult, particularly when considering the rigorous conditions set by formal causal inference frameworks (e.g., those of Pearl, Rubin, or Gelman). These conditions typically require:  1. A priori specification of the ‘true’ causal model structure; 2. Determination of required sample sizes based on the causal structure and assumptions; 3. Independence and random sampling from a well-defined population (or exchangeability, in a Bayesian context); 4. Inclusion or control of all potential confounders (e.g., via the backdoor criterion); 5. No post hoc modification of the model via auxiliary variables or ad-hoc counter-hypotheses is possible. These are exceptionally demanding criteria, and - as acknowledged by leading authors in causal inference
--	---	---

		- they are rarely fully met in empirical ecological studies, particularly those based on large-scale field observations. Our aim, instead, is to estimate plausible values for parameters (β) under the assumption we already had strong a priori information on the magnitude and direction of the relations. The current work is a first step: by quantifying these associations with the available data, we can, in future studies, build on this foundation - using the posterior estimates as priors in a more formal causal framework, generating counterfactual predictions (e.g., removal of one or more stressors), and designing experiments with adequate power.
4	We need a better description of the underlying dataset (a brief overview of time- and spatial-scale of each study, whether they were assess variation along spatial or temporal gradients, typical number of sampling sites / rivers per study etc..)	We went through each individual study again and extracted the required information. We have now added three more tables to the Supplementary Information (Tables S1 and S3), specifying:  - The seasons of sampling - The number of observations per combination of stressor and organism group - The number of datasets per combination of stressor and organism group
5	Statistical analysis: Unclear how the data were grouped to analyse in the glmm. Usually in a synthesis, there is either a two-step approach (producing effect sizes per study and then synthesis of them in a meta-analysis) or a one-step approach (one large synthesis model). Here, we seem to do something in the middle and incorporate both? Rather unclear though.	In contrast to a standard meta-analysis, we extracted the original data from each underlying study and performed GLMMs with each dataset individually. If possible and when it did not lead to convergence issues we fitted random effects e.g. on seasons, periods, years or repeated sampling. This is specified in the supplementary material column of the DATA tab under 'comment' and 'random effect' (https://github.com/snwikajj/Data/blob/main/Unknown_Kajiser_et_al._2025_Supplementary_Information_2.xlsx). With the parameter estimates of the individual GLMMs, we performed a meta analysis. We have now added a section "Overview" in the "Methods": "We systematically identified studies that examined stressor–response relationships under field conditions. For each study, we extracted the data and fitted separate Generalized Linear Mixed Models (GLMMs) on each dataset. The resulting parameter estimates (i.e. regression coefficients) from these individual models were then synthesized through a meta-analysis using Bayesian Model Averaging."
6	You mentioned (e.g., line 276) you estimate the change in the	Thank you for this thoughtful comment. We agree that interpreting a "50% increase" in absolute terms for

responses due to % change in each stressor. But does % increases really make sense for variables like temperature? What would a 50% increase in temperature mean – it means different things across different studies and gradients? I can see why you did this – since the stressors are in different units. But % is a measure of multiplicative change, I am not sure that makes sense for all variables. An alternative approach would have been to normalize the gradients between minimum and maximum values, or to scale in units of standard deviations. Either consider these alternatives or better justify your approach.	variables such as temperature can be problematic - for example, a 50% increase from 10°C versus 20°C implies very different absolute changes. However, our use of percentage change is not based on absolute units but on a log-log or logit-log regression framework, which allows for estimation of elasticities - that is, the proportional change in the response variable per proportional change in a stressor. Here we explain the rationale, and possible alternatives we discarded, in detail, and have opted for a simpler version for the Supplementary Information, which is given below. In a log-log model, the regression coefficient β has a scale-free interpretation: $\log(E(y x_1)) = \beta_1 \cdot \log(x_1)$ $\beta_1 = \frac{\log(E(y x_1))}{\log(x_1)} \text{ simplified as } \beta = \frac{\log(y)}{\log(x)}$ This coefficient represents the elasticity: a 1% change in x leads to a β% change in y. This approach naturally accounts for variables in different units (e.g., °C, mg/L) and provides consistent, interpretable effect estimates across stressors and studies. We considered other scaling approaches - namely, minmax normalization and z-scaling (standardization), but there are significant drawbacks: Minmax scaling is highly sensitive to the specific sample range. Identical values (e.g., 9 mg/L oxygen) could be mapped to very different scaled values across studies with differing ranges (e.g., 8–10 vs. 5–12). This undermines comparability and transferability of the resulting coefficients, because there is no interpretation anymore of the parameter to the outside world. It is a sterile number between -1 and 1. Z-scaling (standardization) scales variables to unit variance but results in coefficients that are difficult to interpret meaningfully. As noted by Greenland et al. (1986) and . Moreover, every value is forced to a unit standard deviation, which limits predictive value beyond the dataset. Both methods obscure the real-world relevance of predictors and are not easily reused in future studies. The log-log or logit-log approach still retains their units under the log scale, the increase in one unit $\log(y)$ unit per one $\log(x)$ unit, which can be used for prediction and display (HOP, PDPs), interpreted and re-used as prior in the next study. Moreover, To further justify our approach, we ran simulations assuming the true model is $\log(y) = -0.3 \cdot x + 3$
---	---

We then applied both minmax and z-scaling to x and y , estimating β across 3,000 simulated datasets. We evaluated bias (deviation from 0.3), variance (spread in deviation), and coverage probabilities of the estimated coefficients. The results showed that:

- Minmax scaling overestimated β and z-scaling consistently underestimated β ,
- Both alternatives had larger variance than the log-log elasticity approach,
- More critically only the log-log method achieved coverage probabilities close to the nominal 95%, while the others dropped to ~70%.

These findings (with full simulation code available on our GitHub repository Link : https://github.com/snwikaij/Data/blob/main/Unknown_Kaijser_et_al._2025_Stochastic_Simulates_Response_1.xlsx) demonstrate the statistical and practical advantages of using elasticities:

1. Effect estimates are interpretable and transferable across scales and units,
2. They retain predictive value, e.g., for future forecasting or management applications,
3. They can be re-used as priors in a Bayesian framework,
4. They maintain higher precision and appropriate coverage under frequentist assumptions.

In conclusion, we acknowledge the concerns about raw percent changes but emphasize that our approach is not based on naive nor non-interpretable proportional increases. It is grounded in a well-established, statistically concept of elasticity via log-

		log or logit-log regression. We believe it offers the most rigorous and interpretable solution. In the Supplementary Information we limited the above explanations to the following: “The elasticity coefficient allows us to express the percent (%) change in the response variable per percent (%) change in the stressor³. For example, an elasticity coefficient of 0.2 indicates a 0.2% increase in the response per 1% increase of the stressor. This approach eased the interpretation between and within groups. Since families and species exhibit similar responses, taxonomic level selection does not affect β_1, facilitating stressor-response comparisons. Furthermore, priors can be easily formulated based on comparison to extremely strong relations. Finally, the ratio $\log(y \text{ unit})/\log(x \text{ unit})$ maintains unit interpretability on the log scale, allowing for consistent comparisons across studies. In contrast, we avoided z-score and min-max transformations because they eliminate unit interpretability and introduce scaling-dependent distortions, which can increase model error (Greenland, Schlesselman, and Criqui 1986). Which means that the model can still be compared beyond the study while after scaling this is impossible. These transformations hinder comparability across studies. For example, with z-standardization, “coefficient estimates may not necessarily be comparable across studies that use different samples.” (Penney 2017), since standardization depends on sample-specific means and standard deviations. To illustrate, suppose Study A has a mean of 500 and standard deviation (SD) of 300, and Study B has a mean of 150 and SD of 50. If a new observation is 300, then under z-standardization: $\text{Study A: } = \frac{(300-500)}{300} = -0.67$ $\text{Study B: } = \frac{(300 - 150)}{50} = 3$ Despite the raw value being identical, the standardized values differ drastically, introducing variability and reducing precision in parameter estimation. Min-max normalization similarly assumes a consistent data range across studies. For example, for oxygen measured in mg/L: $\text{Study A: } 5 - 12 \text{ mg} \cdot \text{L}^{-1} = \frac{(9-5)}{12-5} = 0.57$ $\text{Study A: } 8 - 12 \text{ mg} \cdot \text{L}^{-1} = \frac{(9 - 8)}{12 - 8} = 0.25$
--	--	---

		Again, the same raw value (9 mg/L) is mapped to different scaled values, resulting in non-comparable model parameters. In contrast, $\log\text{-log } \beta_1 = \frac{\log(y)}{\log(x)}$ or $\text{logit-log } \beta_1 = \frac{\text{logit}(y)}{\log(x)}$ transformations are scale-invariant and preserve the structure and units of the original data, making model estimates more interpretable and comparable across different datasets.”
7	The results focus on the results in binary terms – positive/negative – I think it would be more interesting to focus on the relative magnitudes of the stressors, since this is the key area of uncertainty (this comparison could be aided by the same axes scale in Fig. 2 or in a different figure). Also, as in most meta-analysis, it would be nice to report measures of heterogeneity - i.e., are the relationships more variable for some stressors than others? Currently, the focus is rather on the mean effect only, but there is a lot more information here potentially, relevant for interpretation.	Thank you for this constructive suggestion. We fully agree that focusing on the relative magnitudes of stressor-response relations, as well as the variability (heterogeneity) in these relationships, provides insight and enhances the interpretability of our findings. Regarding the magnitude of stressor-response relations, we intentionally chose to simplify the main figures to reflect the directionality (positive/negative) for a broader readership, in line with the general style of Nature Ecology & Evolution. However, we recognize the importance of stressor-response relations and heterogeneity, and thus we have provided the full posterior summaries, including mean, MAP (maximum a posteriori estimate), standard error, 90% HDI (highest density interval), I^2 (as a measure of heterogeneity), and sample size per stressor, in the Supplementary Information SI3 and on GitHub (link: https://github.com/snwikajj/Data/blob/main/Unknown_Kajiser et al. 2025 Supplementary Information 3.xlsx). To clarify, heterogeneity estimates (I^2) in our analysis span a wide range - from near 0% to over 99% - depending on the stressor. We acknowledge that this information was underemphasized in the main text, and have now addressed it with the following paragraph (Results and Discussion, 3rd paragraph): “In many individual studies, the relationships between stressors and biological responses varied greatly, often displaying substantial heterogeneity. Although some stressor-response patterns are evident, the overall relations are relatively modest and highly variable for different stressors and organism groups. This highlights the inherent complexity of interpreting stressor-response dynamics in ecological systems.”
8	Line 34: potentially misleading – here you report 22,120 studies but later only 277 (line 93).	Thank you. Procedure and number of studies / stressor-response relationships at different levels should be obvious from Figure 5, however, we agree that the abstract must be understandable “stand alone”. We have therefore modified the text as: “We

		screened 22,120 papers and extracted 277 studies with 1,334 stressor-response relationships”.
9	Line 46: I am not sure this blanket statement is fair and representative: freshwater insects have done better than many others e.g., Haase et al. 2023 Nature; van Klink et al. 2020 Science. Some nuance is needed here.	We agree and have changed the sentence from “the most rapid and severe biodiversity declines of all biomes” to “rapid and severe biodiversity decline”. This is still fair to say - but the superlative seems less justified, also in light of the recent paper by Keck et al. (2025) in Nature.
10	Line 77 – since this nature paper just came out (The global human impact on biodiversity Nature), you’ll also need to distinguish yourself from that.	In the paragraph above, we have now added: “While there are multiple studies on individual stressor-response relations (Orr et al. 2024) and a general overview on differences between terrestrial, marine and freshwater communities (Keck et al. 2025), an aggregated quantification of how aquatic organism groups are related to a range of stressors is yet missing.”
11	Line 79 – make clear whether you include experimental and/or observation data here	We have now added: “Drawing from 22,120 observational studies”. For a justification of the limitation to observational studies, please compare our response to Comment 3.
12	Line 79 - in fact, I’d say here include the specific set of inclusion criteria that you used to determine the relevance of a study. Also some sort of statement on how you made the results comparable across different studies (presumably they varied in the magnitude of their stressor gradients. I appreciate its an NEE paper, but a bit more on methods is needed, but the results can be interpreted.	We have now included some general information on the studies that have been selected, i.e. that we limit our selection to observational studies and did not include experimental studies. We have also extended the section “Data collection and literature search” in the “Methods” chapter. All details on how we selected studies are specified in the Supplementary Materials. Concerning the required statement on how results were made comparable across studies (see also our response to comment 6): We used a log-log or logit-log regression framework, which allows for estimation of elasticities - that is, the proportional change in the response variable per proportional change in a stressor. In the first chapter of the manuscript (paragraph 5), we have now added: “Hereby, we used a regression framework that links the proportional changes in stressors and responses.”
13	Line 100 – since this is the largest pool, report the % for richness and abundance separately.	It has been changed to: “Taxonomic richness (e.g., species or genus counts) that were analyzed with log-linear models prevailed (92.9%), while for all logit-linear models evenness prevailed (71%) (Table S1).”

14	Line 257 – so you kept both field experiments and observational field studies. Any difference between these should be tested since each typically vary in internal vs external validity. For observation field studies, was there any attempt to control for confounders when assessing the impact of any specific driver? Was the gradient a spatial gradient (e.g., sites in different temperatures) or a temporal gradient (e.g., temperatures vary among years). The latter is more likely to be a causal in most observational studies since spatial gradients of different variables typically covary with each other.	We apologize for this misunderstanding. In line 257 (referring to the original manuscript version) we stated: “To ensure ecological realism, we excluded laboratory-based experiments, restricting our analysis to field studies.” Therefore, there was no need to correct for field studies vs. experiments in our models. We did not consider confounding variables, as it was not possible to extract those in a harmonised way from the multitude of studies. Please also note that we did not attempt to make any causal claims - the study is limited to the description of stressor-response relationships (compare also our response to comment 3).
15	Line 265 – I expect some information here on data extraction – i.e. what information did you extract from each study. Its unclear at this stage whether you were seeking essentially the raw data from the studies, or whether you wished to directly extract coefficients for the effects of different stressors from the studies, or summary variables.	Thank you. We have clarified this now as follows: “We extracted data on key response variables, including taxonomic richness and evenness, from figures, tables, and supplementary datasets (...).”
16	Line 269 – as mentioned above, I am unclear what a dataset would look like. Is it e.g., species richness under different values of a stressor gradient? Did you include studies treating both stressor as a factor versus a continuous variable? Spatial and temporal gradients?	Thank you for uncovering this confusion. In fact, we used the term “dataset” for both, the entire datasource we extracted from literature, and for the individual stressor-response relationships stemming from individual studies. In the revised manuscript, we now use the term “individual dataset”, defined as an individual stressor-response relationship stemming from a single study. We limited the selection of the underlying studies to those that display continuous variables and omitted those using factors. Spatial and temporal gradients are included as random effects, if they could be retrieved.

		To clarify this, we have added the following explanations in the section “Data collection and literature search”: “As independent variables, we extracted proxies of stressor intensities (for a full list of proxies please compare Supplementary Information, Step 1). We also extracted if the study was based on a temporal or a spatial gradient (see Fig. 6, Step 1, and Supplementary Information 1 Step 1 for details). We refer to the data extracted from an individual study as “individual dataset” in the following.”
17	Line 269 – not sure what ‘dataset’ refers to here. Is it per study? Or per taxon/response group? Or?	See comment 16. We added “individual” dataset.
18	Line 273 – mention what the independent variables were. Did you build separate models for each stressor? Or use multiple regression? If a study included ‘control’ variables e.g. sampling effort, did you also include those?	The full list of independent variables is given in the Supplementary Information, Step 1. We refer to this now in the section “Data collection and literature search”: “As independent variables, we extracted proxies of stressor intensities (for a full list of proxies please compare Supplementary Information, Step 1).” We build separate models on each individual dataset, i.e. on each individual stressor-response relationship stemming from a single study. We have added this information now in the section “Model fitting and parameter extraction”. In the original manuscript version, random effects were only mentioned in the “Supplementary Information”. Now we have included in the main manuscript (section “Model fitting and parameter extraction”): “If studies provided information on sampling dates, seasons or years, or on individual rivers sampled, random effects were applied if they did not lead to convergence issues of the model.”

19	Line 287 – Not sure what you mean by ‘accumulation’. Synthesis? But that would be confusing since you already have a later section on meta-analysis.	We agree that the term “accumulation” is misleading and have replaced it by “storage”. The paragraph reads now: “Storage of estimated parameters” All the estimated elasticity and semi-elasticity coefficients (regression coefficients, intercepts) for each stressor-response relationship were stored in a database (Fig. 6, Step 3; Supplementary Information 1, Fig. S2b, Step 3).”
20	Line 321 – since you used quite strong priors, this sensitivity analysis using uninformative priors is important. Here you are reporting the results? I don’t follow what you mean. The results were the same or? Either way, mention in the main text.	Thank you. We have added in the Methods Chapter (Step 8: Posterior sensitivity check): “We conducted a sensitivity analysis to evaluate the extent to which the priors influence the posterior estimates. To do this, we compared the posterior results presented in the main text with those from an alternative model using diffuse priors (i.e., priors without specific prior information) (Fig. 6, Step 8). The analysis revealed that our prior assumptions about the stressor-response relationship tend to be more negative than the estimates derived from the data. Although some deviations from zero were observed, the estimated mode (i.e., the central tendency) remained stable (see Supplementary Information 1, Step 8).“
21	I’d rather have Fig 1 as a panel plot – you show geographic coverage, but I’d include for b. taxonomic coverage and c. stressor coverage across the studies. (Table 1 could then be moved to SI – especially for NEE, I’d go for plots than tables for easier communication).	We have amended the figure by a panel plot organism groups per stressor. Table 1 has been moved to the Supplementary Material.
22	Figure 4 – the models assumed additive effects? In that case, I don’t think it makes sense to show 2-D covariate plots.	This is one of the few comments with which we respectfully disagree. Partial dependence plots (PDPs) are commonly used to visualize additive relationships - see, for example, Fig. 4B in Shiroyama and Yoshimura (2016) or Fig. 12B in Badr, Zaitchik, and Guikema (2014). Additive effects can vary in magnitude, which in turn affects the diagonal pattern and the rate (or intensity) of change, as reflected in the colour gradient. Therefore, PDPs can effectively

		represent additive results as well. For this reason, we have chosen to retain Figure 4.
23	Figure 5 -I don't think its so helpful to specify the number per model type (log-linear or logit) as per bottom row. Of more interest biologically, is the number of each response variable.	We have added a line in the lower part of the diagram specifying the number of stressor-response relations per organism group for log and logit linear models.
24	Figure 6 showing the analytical workflow is important but the diagrams are not informative enough. I suggest adding more text to explain the steps (and use the step numbers in the text more).	We have now added short statements in the diagram on what the individual steps are about: Step 1: Extraction of data on stressor-response relationships Step 2: Model fitting and parameter extraction Step 3: Storage of estimated parameters Step 4: A priori bias assessment Step 5: Generation of four priors for each stressor-response relationship Step 6: Meta-analysis with regression coefficients Step 7: Posterior bias check Step 8: Posterior sensitivity check Step 9: Visualization
25	Line 86 – I am not sure what you mean here by false-negative rates nor how it was assessed. This whole para needs more detail on rationale and methods.	(This comment refers to Supplementary Information 1) We have amended the paragraph as follows: “Prior to article screening, false-negative rates were assessed using a pre-screened set of 33 articles (14 containing relevant data), which was provided to each analyst to check his/her ability to detect the presence of extractable data from figures, tables and supplements. The false-negative rate, i.e. the rate of relevant articles the analysts overlooked, was 0.23 (...)”.
26	Line 88 – who did the analysts receive feedback from? Each other?	(This comment refers to Supplementary Information 1) The feedback was given by the first author. We have added this information in the text.

27	Line 96 – I would separate inclusion criteria (what determines whether a study was included or not) and data extraction (what was extracted from included studies) into separate sections	(This comment refers to Supplementary Information 1) We have separated information on inclusion criteria and on data extraction into different paragraphs.
28	Line 101 – what biodiversity metrics were of interest? Also mention here the inclusion criteria on study design and stressor types of interest.	(This comment refers to Supplementary Information 1) We have added: “Studies were only considered if biodiversity metrics were reported (evenness, Shannon diversity, number of species, genus, family or order, abundance).” Concerning study design, we added: “It [the search string] was designed to exclude studies on laboratory experiments. Any laboratory studies were discarded.” Concerning stressor types of interest: These are listed in the 3rd and 4th line of the section “Step 1: data sources and systematic search”.
29	Line 111-113 – do you mean to imply that %EPT-taxa and coverage% were treated as evenness metrics. How do you justify this?	(This comment refers to Supplementary Information 1) We have re-written the paragraph to explain that logit-linear models were used for all “proportional metrics”. These include evenness, but also cover (in case of macrophytes) and percentage of EPT taxa (in case of invertebrates).
30	Line 103 – since we haven’t started talking about models at this point, I’d just frame this section as response variables of interest – the distinction between logit and log should come into the next section about model fitting.	(This comment refers to Supplementary Information 1) We removed all reference to logit- or logit-linear models from the paragraph.
31	Line 106 – the main text also mentioned abundance (line 100)	(This comment refers to Supplementary Information 1) We removed abundance from the main text for the log linear models. This was a mistake that we have now corrected.
32	Line 116 – but how does such a categorization help assess bias? And in any case what sort of bias are we meaning? I think	(This comment refers to Supplementary Information 1) We apologize for this lack of clarity. In general the main text and figures often report the strongest and most

	this sentence needs to be written either way.	obvious patterns. We have amended the text as follows: “Data sources were categorized as “figure,” “table,” or “dataset” to assess bias, i.e. if stronger stressor-response relationships were observed for data presented in figures and tables compared to data given in the annexes (...).”
33	Line 119 - is this fitting models to each dataset/study separately? Or are we already doing synthesis (line 81 in your main text indicates that the glmms and meta-analysis were two separate things, but now I am not sure)	(This comment refers to Supplementary Information 1) Thank you for spotting this inconsistency. For “Step 2: Model fitting” we have now added: “To each individual dataset, we fitted a employed Generalized Linear (Mixed) Models (GLM(M)s) with log- or logit-links (...).”
34	Line 126 – obviously seasonal effects vary across the globe, but you are modelling it as a constant variation here as a random effect - whether your approach is valid here depends on whether these are separate models for each dataset/study or not? If these are already synthetic models, the seasonal effect could be allowed to vary among studies/regions (actually at this point, I realise I am unclear on how long these studies typically are – annual long-term or short-term daily measurement studies? Some description of this is needed, also how many sampling sites/rivers does each study tend to have?).	(This comment refers to Supplementary Information 1) We agree that seasonal effects may greatly vary between studies. In fact, we have built a GL(M)M for each individual data set and included seasons as random effects if feasible. See our response to comment 33. For “Step 2: Model fitting and parameter extraction” we have now added: “To each individual dataset, we fitted a employed Generalized Linear (Mixed) Models (GLM(M)s) with log- or logit-links (...).”
35	Lines 129-131 comes as a surprise since we seem to be modelling all variables at the same time. This was not clear before. So was another inclusion criteria that a study measured multiple stressors? Why isn't dataset or study mentioned as a random effect? As you might be gathering by	(This comment refers to Supplementary Information 1) We are sorry that we didn't explain our approach sufficiently in the previous manuscript version. In fact, models have been fitted on each individual dataset after which parameters were extracted from the models in Step 2 (see for example our responses to comments 33 and 34).

	now, I am rather confused by the models!	So, the lines that you are referring to were describing our approach towards the individual GL(M)Ms that were fitted on each dataset. They are NOT describing the subsequent meta analysis, which is specified in Step 6. We hope that the overview we have added at the beginning of the Methods Chapter in the main manuscripts contributes to solving this confusion. In the Supplementary Information, we have added at the beginning of Step 2: “To each individual dataset, we fitted a GLMM...”.
36	Line 153 – what does accumulation mean? Synthesis? Aggregation? Not sure what is happening in this step. Is it processing the model outputs?	(This comment refers to Supplementary Information 1) We have replaced the headline by “Extraction of estimates” (compare also comment 19)
37	Line 167 – what sort of bias? Publication bias?	(This comment refers to Supplementary Information 1) Yes, we addressed publication bias, to assess if data extracted from literature favour certain response categories, e.g. stronger over weaker responses. We have added this information under Step 4 of the Supplementary Information.
38	Line 173 – so you have multiple values per study - I am still unclear what they are and what you are aggregating in the same model here. Also - why would source (figure or table) be relevant?	(This comment refers to Supplementary Information 1) Each model fitted can refer to between one and seven stressors plus an intercept. Each stressor-response relation is separately modelled in the random effect-meta analysis (compare also the answer to comment 35). The source (figure, table, dataset) is potentially relevant as we aimed to test if stronger stressor-response relationships were observed for data presented in figures and tables compared to data given in the annexes. We have added this information to the Supplementary Information.
39 check	Line 193 – according to what assumptions should this be normally distributed-explain the thinking more.	(This comment refers to Supplementary Information 1) We added the following short explanation: “In short, since the estimate could, in theory, take any value from $-\infty$ to ∞ and the standard error (SE) must be greater than 0, the possible values for $z=\text{estimate}/\text{SE}$ also range from $-\infty$ to ∞.”

		According to the central limit theorem, as the sample size within each study increases, the sampling distribution of the individual study estimates tends to approximate a normal distribution. Thus, we can model the estimate as approximately following a normal distribution. $N(\mu, SE)$, where μ represents the (unknown, fixed or random) underlying population mean and SE is the standard error of the estimate. When we standardize the estimate by subtracting μ and dividing by its standard error, we obtain the standardized statistic: $Z = (\text{estimate} - \mu) / SE$. If we assume $\mu = 0$ then it simplifies to $Z = \text{estimate} / SE$. Under these assumptions, the Z-statistic approximately follows a standard normal distribution.”
40	Line 201 – we are talking about priors before talking about what any Bayesian model would seek to do... Is this about the earlier models (if so it should be moved there, but it wasn't mentioned that the GLMMs would be fit within a Bayesian framework) or the next models for the meta-analysis?	(This comment refers to Supplementary Information 1) Any Bayesian model seeks to incorporate prior information. Accordingly, a Bayesian meta-analysis requires the definition of priors before the analysis is performed. We have added the following information to the Supplementary Information (Step 5) to clarify our approach: “A key element of our approach was the application of Bayesian Model Averaging (BMA) that allows for guiding models toward plausible stressor-response relationships based on prior beliefs⁵⁵. BMA requires the generation of priors.” This is the same text we have now also use to justify Step 5 in the main manuscript.
41	Line 223 – write an opening section about what you seek to do with these priors? Are you trying to integrate existing knowledge? Given you have a lot of data, I would have suspected that the priors are less important here, so I am unclear why you take a more complex approach than many similar syntheses (that usually use uninformative priors). Again, some rationale sentences would be helpful.	(This comment refers to Supplementary Information 1) Thank you for raising this important point. We agree that the role of priors in Bayesian analysis is often under-discussed, and we appreciate the opportunity to clarify our rationale. The first part of our answer is now also used in the Supplementary Material (Step 5: Prior formulation, section “Prior specifications”). Our use of priors serves two main purposes:  1. To incorporate a range of plausible ecological expectations, based on existing notions of "diffuse", "ignorant", "directional", and "strong" informative priors. 2. To adhere to the principles of Bayesian inference, where not specifying priors is essentially neglecting the core tenet of Bayes'

		theorem—that prior information should inform the analysis. We did not rely on a single prior. Instead, we explored four different priors, ranging from weakly to moderately informative:  • A Diffuse prior (uninformative prior) Normal(0, 10) • A Ignorant prior Normal(0, 0.5) to reflect minimal prior knowledge only about its range. • An informative prior Normal(-0.3, 0.3) or N(0.3, 0.3) that reflects widespread ecological expectations - for example, the notion that increasing salinity tends to reduce freshwater invertebrate diversity at broad scales. • A Strong prior Normal(-0.2, 0.1) and N(0.2, 0.1) that reflects widespread ecological beliefs and salinity tends to reduce freshwater invertebrate diversity at broad scales. Our approach does not constrain the analysis to either positive or negative stressor-response relationships. Rather, it embraces a range of ecologically plausible scenarios and allows the data to update and refine this prior information. Using Bayesian model averaging, we assess the contribution of each prior to the posterior distribution. This is done under the assumption of equal but stochastic prior weights, modeled via a Dirichlet distribution. This framework enables us to identify which priors are most consistent with the observed data. The following explanations have not been used for the manuscript) Running the model it is possible to check how plausible the data is under an uninformative prior divided by how plausible the data is under a directional or strong prior. $BF = \frac{P(Diffuse\ prior Data)}{P(Informative\ prior, Strong\ prior Data) \cdot \frac{P(Diffuse\ prior)}{P(informative\ prior, strong\ prior)}}$ For the sum of individual stressor-response relation together #Extract the posterior simulated odds
--	--	--

		<pre> bf <- mod\$Chains_podd[!(mod\$Chains_podd\$predictor) == "NA",] #Display the simulate per prior 1=Diffuse prior, 2=Ignorant prior, #3=Directional prior, 4=Strong prior table(bf\$estimate) #Probability of the data favoring the Diffuse prior over the Directional or strong prior plogis(log(table(bf\$estimate)[1]/sum(table(bf\$estimate) e)[3:4]))) #Probability of the data to favour the N(0, 10) is only 0.45% #0.0045=P(Data Diffuse prior)/P(Data Informative prior, Strong prior) Therefore any prior N(-0.3, 0.3), N(0.3, 0.3), N(-0.3, 0.15) or N(0.3, 0.15) is more plausible than an uninformative prior (diffuse prior). Using an uninformative prior was already known to be unreasonable and has now also been justified by referent to the posterior odds. This can also be assessed for each individual stressor- response relation where fractions of the simulations from the Directional or Strong priors are more probable. The stochastic BayesFactors this consistency informs us that these priors are most plausible. It is possible to examine this by dividing the fraction of simulates from a particular prior that end up in the posterior by each other provided in the posterior odds table (https://github.com/snwikaij/Data/blob/main/Unknown_Kaijser_et_al._2025_Stochastic_Simulates_Response_2.xlsx). For example the diffuse (uninformative prior) is hardly selected for invertebrates salinity and a log-linear model. BF = 0.824/0 while for the ignorant prior this is BF = 0.824/0.06 = 13.7. In short, our use of priors is not intended to strongly steer the results, but to ensure transparency, test robustness, and embed existing ecological understanding into the statistical framework in a principled way. </pre>
--	--	--

Reviewer #2

	Comment	Response
42	In general, this is an interesting and quite unique contribution that seeks to identify broad relationships between putative stressors and different taxonomic groups of freshwater organisms at a global scale. I think the work will attract broad interest, but needs:	Thank you!
43	i) Some attention to the clarity of the writing in places (expanded below)	We have now modified the manuscript in many places to enhance clarity. For details please see our response to comments 47 to 66.
44	ii) Recognition that some important potential stressors have not been included – for example invasive non-native species and emerging contaminants	Thank you for this important observation. Please compare our response to comment 54.
45	iii) Recognition that some pressures on freshwater organisms could arise from life stages in other systems – for example marine ecosystems in the case of migrant fishes or terrestrial systems in the case on insects with flying adult stages	Thank you and please compare our response to comment 57.
46	iv) Some more clarity with respect to the nature of the relationships detected – for example whether they are linear, curvilinear etc	Please compare, for instance, our response to comment 60.
47	I think there are likely to be taxonomic challenges in understanding the data. While biodiversity is relatively straightforward to appraise in, say, fishes or macrophytes, this is not so easy in groups such as bacteria or archaea – where identification depends on features such as gene	We agree. This is not only a consequence of identification methodology (field identification vs. microscopic identification vs. barcoding) but also a matter of how mature the taxonomy is. Even the species concept is not necessarily comparable between bacteria/archaea and the other taxonomic groups. To account for this, we have now added in the section “Macro- and microorganisms exhibit

	sequences or even gene expression	divergent relations to stressors” (second paragraph): “The response of bacteria/archaea is often divergent compared to macroorganisms, likely due to their environmental specificity and the underrepresentation of relevant studies. This includes methodological challenges to designate species and to assign operational taxonomic units that respond to stress intensities. ”
48	Even with relatively large data sets (as here) – I wondered also about intrinsic biases in the global data available? Examples are where invertebrates or fishes are likely to more readily sampled (and therefore more often studied) from lower-order river systems than from large rivers – where stressor combinations and effects could change.	We agree and have addressed biases in various steps of our analytical procedure, in particular in Step 4 (where we assessed if data extracted from literature favour certain response categories, e.g. stronger over weaker responses) and Step 7 (posterior bias check). Still, these steps do not account for various other sources of bias such as stronger representation of certain countries, river types, stressors or organism groups in the data set. We have now added in the section “Research and management implications” (second sentence): “First, the data used in this study reflect the research priorities of recent decades and are not necessarily representative of the actual relevance of stressors, organism groups, or river types. The dataset is heavily dominated by studies from a few countries - particularly the United States, China, and Germany - and macroinvertebrates are substantially better represented than other organism groups.”
49	I also wonder what may detail may be masked at the level of taxonomic aggregation used here. The prime example would be the lumping, say, of bryophytes and angiosperms as ‘macrophytes’ – but these two have very different evolutionary origins and environmental requirements. Similarly, responses to pressures in different sub-groups of the other taxa (fishes, invertebrates...) are likely to vary strongly with adaptation in ways that cannot be captured with such aggregated analysis.	We agree that this is a potentially important source of information loss. However, one of the primary purposes of models - and meta-analyses in particular - is to synthesize and condense complex information, which inevitably involves a trade-off with detail. Even at the level of the metrics we employ (e.g., species richness, evenness), substantial information about individual species’ preferences and ecological roles is lost. We believe that detailed analyses of how specific sub-groups or individual species respond to stressors lie beyond the scope of a meta-analysis and are more appropriately addressed in the original studies on which it builds. Moreover, our analysis is constrained by the level of detail reported in the primary literature. For instance, most studies referring to fish or macrophytes do not distinguish between functional groups such as bryophytes and vascular plants. While a separate

		analysis for bryophytes would indeed be valuable, the available data do not support such a resolution. To acknowledge this limitation, we have added the following sentence to the section “Macro- and microorganisms exhibit divergent relations to stressors” (end of first paragraph): “The meta-analysis necessarily obscures divergent responses of species groups within an organism group, such as bryophytes and vascular plants.”
50	Lines 32, 33 and 37: could you indicate which aspects of biodiversity you addressed? Richness? Alpha? Beta...?	We have now added in the abstract: “Consistently, elevated salinity, oxygen depletion, and fine sediment accumulation were negatively related to biodiversity (taxa richness, evenness) across taxa, (...)”. More details are given in the Methods chapter (Step 1) and in the Supplementary Information 1 (Step 1, section “response variables”).
51	Lines 39/40: The multiple adjective makes this slightly hard to read (critical, multivariable, stressor-response. And tailored management strategies for what purpose?	We agree this sentence included a certain amount of waffle. The revised version reads: “Predictive tools, including Hypothetical Outcome Plots and Partial Dependence Plots, revealed the interplay of stressors and predicted biodiversity response to stress increase.”
52	Line 43: can you genuinely extrapolate from your analysis to an understanding of resilience?	We have removed the reference to resilience. The sentence now reads: “(…) and informing conservation strategies for freshwater ecosystems.”
53	Line 48: this feels in need of some expression of temporal as well as spatial scales: global biodiversity decline is a process through time.	We agree and have rewritten the first sentences as follows: “Freshwater ecosystems - particularly rivers - are experiencing rapid and severe biodiversity declines. These declines are driven by multiple interacting stressors operating across local, catchment, and global scales, with global biodiversity loss manifesting over recent decades ^{1,2}.”
54	Line 47-55. I respect that these are examples of global change pressures – but I wonder at the global effect of invasive non-native species or the overlooked effects of emerging	It is correct that we have been selective. We focussed on stressors that we (i) well documented with plenty of studies addressing them and (ii) that are straightforward to parameterize. Parametrization is particularly challenging for

	contaminants? These are poorly parameterised in many studies. Perhaps a point for discussion.	contaminants that include many substances, and for invasive species. We have now added to the discussion (section “Research and management implications”): “In addition, we did not consider various emergent stressors, e.g. contaminants and invasive species, that are more challenging to parameterize.”
55	Lines 58 and 59: should these numbers be reflected by the names of the authors responsible?	We have added the authors’ names.
56	Lines 66-68: what of biotic stressors – where species interactions are involved? There is some evidence that biodiversity decline can be mediated by interactions between global change processes and predator/prey or competitive relationships. (eg https://www.journals.uchicago.edu/doi/10.1899/09-159.1)	It is certainly true that biotic interactions could greatly mediate responses to stressors, and this may particularly concern interactions with invasive species. However, we believe that this is beyond the scope of a meta-analysis. We are referring to the changes we have implemented following comments 49 and 54.
57	Line 73: there are some complications, for example where fishes are long-distance migrants that move between realms or where insects have aerial stages that interact with terrestrial ecosystems.	We agree and have modified the sentence as follows: “Larger organisms, such as fish and macrophytes, are disproportionately affected by habitat modifications including associated dispersal constraints ^{20,21} , while (...)”.
58	Line 78/79: there are some taxonomic challenges in understanding the focal units being assessed – for example between bacteria versus fish	This is an important limitation of our study. We have addressed this now through the modifications we have implemented following your comments 47, 48 and 49.
59	Line 110: and the difficulty of understanding ‘taxonomy’ in micro-organisms?	Please also compare our response to comment 47. We have now added in the section “Macro- and microorganisms exhibit divergent relations to stressors” (second paragraph): “The response of bacteria/archaea is often divergent compared to macroorganisms, likely due

		to their environmental specificity and the underrepresentation of relevant studies. This includes methodological challenges to designate species and to assign operational taxonomic units that respond to stress intensities. ”
60	Line 124-134: this paragraph could be stronger. (i) First, highly saline marine ecosystems have rich algal communities – so the effects of salinity may not be so straightforward. (ii) Second, the nature of the relationship between nutrients and algal richness is not made clear: is it linear or curvilinear? (iii) Third, lines 132-134 illustrate the challenges of assessing diversity in different groups where methods are not standardised or assessments made in ways that don't account for sample size (eg through rarefaction)	Thank you for these helpful observations. We have revised the paragraph to better reflect the complexity of these relationships: (i) The second sentence of the paragraph has been modified as follows: “The negative relationship to salinity was particularly strong, likely driven by osmotic stress ^{18,36}, supporting the conjecture that freshwater species are not directly replaced by brackish water species when salinity increases.” (ii) In fact, these relationships are curve-linear. We have added this information to the first sentence of the paragraph. (iii) We agree (though these challenges have been addressed by using elasticity coefficients in our analysis) and hope that the sentence reflects these limitations adequately.
61	Line 133-134: the logic here could be clearer	We have amended this sentence as follows: “Other stressor-response relations were weaker, suggesting indirect relations (e.g. through another environmental variable) and data insufficiency.”
62	Line 135-140: How are ‘macrophytes’ defined? It's likely that the patterns described might differ, say, between bryophytes and angiosperms	We have added that macrophytes include bryophytes and vascular plants. Compare also our response to comment 49. To acknowledge this limitation, we have added the following sentence to the section “Macro- and microorganisms exhibit divergent relations to stressors” (end of first paragraph): “The meta-analysis necessarily obscures divergent responses of species groups within an organism group, such as bryophytes and vascular plants.”
63	Line 141 and 142/143: these lines clearly overlap	We have merged these two sentences into one.

64	Line 144-146: the logic could be clearer: why does this imply an indirect relationship? And in what sense?	We have amended the sentence as follows: "Nutrient enrichment caused a weak response, suggesting that direct nutrient impacts may be less relevant than habitat alterations and acts in an indirect way, e.g. through temporary oxygen depletion ¹⁹ .
65	Line 149-151: this contrast with invertebrates is interesting: does it have evolutionary origins? Fishes colonised freshwaters from marine environments while many invertebrates (especially insects) colonised freshwaters from terrestrial realms.	This is an interesting consideration. However, we are reluctant to add this thought to the manuscript as we can not directly test it with our data.
66	Line 176: threat	Corrected

Reviewer #3

67	This synthesis presents some very interesting and relevant data that would be of interest to a broad readership and could have important implications for informing management directions of river systems. To achieve this, however, the descriptions and discussion of the results require some work to tell a cohesive story that is fully supported by the data. I encourage the authors to consider restructuring the discussion of their results around the key stressors of interest, rather than paragraphs summarising responses of each taxonomic group. Exploring some of the strongest stressor responses in more detail (as shown for the selected HOPs and PDPs, particularly for important stressor interactions such as nutrients x warming, sedimentation x flow velocity, for example) and highlighting key management	Thank you! Following your (and the other reviewers') suggestions we have modified the manuscript in many places. For details, please compare below.
----	--	--

	implications that result from these analyses would considerably enhance the value of this paper. With some revision, there is certainly potential for an interesting and informative piece that provides important insights for conservation strategies and highlights the potential for future analyses utilising similar approaches, as per the concluding sentence of the abstract. I have provided specific comments below that I hope will be of use in improving the manuscript.	
68	L55: Here and throughout, I personally find the use of the term “relations” a bit strange, to me this is more a term used with respect to people, e.g. family, and suggest “relationships” which, while almost the same word, I am more familiar with in scientific terms. Elsewhere I have suggested using “responses”, instead of relations.	Thank you for this observation. We acknowledge that our use of the term “relations” to describe the link between stressors and biota may be unconventional, particularly in English, where “relations” is often associated with interpersonal or familial contexts. Our choice was deliberate, however: we intentionally sought to avoid terminology that implies causality, such as “effects” or “responses.” We have elaborated on this rationale in our response to Comment 3 from Reviewer #1. In short, terms like “effect” and “response” often carry causal connotations, which we believe should be used with caution - especially in meta-analyses like ours, where the assumptions required for formal causal inference (as outlined in the frameworks of Pearl or Rubin, and Gelman) are rarely satisfied. While this terminology was initially debated among the co-authors, we ultimately agreed that a more cautious, epistemologically informed wording is appropriate for the nature of our study. We have since consulted several native English speakers, who confirmed that although “relations” may sound slightly unusual in this context, it is grammatically correct and not misleading. On that basis, and given our intent to avoid overstatement of causal interpretation, we have opted to retain the term.
69	L61: Suggest altering “aimed at unravelling” to just “investigated”.	We have replaced “aimed at unravelling” by “investigated”.
70	L62-3: Possibly something briefly discussing the challenges of	We have added:

	developing such a predictive framework would be useful.	“This would require information, ideally raw data, on the relation between various organism groups and stressors from a range of ecoregions and river types, and a model that accounts for these multivariable data and the associated bias.”
71	L64-6: It is not clear to me why this is the case. Orr et al. 2024 showed there are almost 2400 studies investigating the individual and combined effects of multiple stressors, generally on specific organisms? So the meaning behind this paradox is unclear.	Thank you for this important comment. While we agree that there are multiple studies, both from experiments and field observations, on the relation between stressors and biota, they have not yet been aggregated and generalized. This requires at least one continued independent and response variable, respectively, which need to be made comparable across studies. We have added the following sentence: “While there are multiple studies on individual stressor-response relations (Orr et al. 2024) and a general overview on differences between terrestrial, marine and freshwater communities (Keck et al. 2025), an aggregated quantification of how aquatic organism groups are related to a range of stressors is yet missing.”
72	Further comment: I think the confusion for me lies in the use of “specific organism groups”, because I initially read this as taxonomic groups at e.g. species-to-family level. But given the focus of the meta-analysis on broad groups as described in L77-9, I think this is really what the authors are getting at with respect to the paradox - that the focus on specific taxonomic groups has obscured the interpretation of absolute effects on broader groups?	To a certain degree, this interpretation is correct. Many of the individual studies, e.g. those referred to in Orr et al. (2024), are limited to more specific organism groups. However, we think that the main gaps are in the lack of aggregation of individual studies and use of relative or standardized effect-sizes (Bagueley, 2009), which limits the quantification of how organism groups are related to stress intensities. We hope this is now clearer given the changes we have implemented based on comment 71.
73	L66-8: What about biological stressors? Invasive species, pathogens, biotoxins, etc.	We agree that we have been selective, but the selection was not “cherry picking”, but demanded by our analytical approach. We could only use stressors that can be condensed into a predictor that is common across studies (such as phosphorus concentration or temperature change). The impact of invasive species, for instance, is hard to channel into a single predictor. We have now added to the discussion (section “Research and management implications”):

		“In addition, we did not consider various emergent stressors, e.g. contaminants and invasive species, that are more challenging to parameterize within a BMA framework.”
74	L68-71: “Each stressor operates through distinct mechanisms”; However, these mechanisms are not always known or well understood. Moreover, there are many different levels of these mechanisms - the example of fine sediment accumulation or channelisation altering habitat availability is a very broad mechanism which has many sub-mechanisms that will result in which species are favoured vs excluded. These are quite different to the specific cellular mechanisms of osmoregulation and slowed metabolism mentioned in the previous sentence. Possibly some brief discussion of this issue could be useful (i.e. that the mechanisms of operation/mode of action of stressors can operate at different levels, and how some stressors operate are more well understood than others at different levels - from the cellular to habitat scale.	We fully agree to this observation and have now amended the text as follows: “Each stressor operates through distinct modes of action that may be caused in specific cellular mechanisms or in the provision/removal of habitats and that exclude or favour certain species.”
75	L74: Suggest “broader” instead of “broad”, Using “while” to begin the second half of the sentence suggests a comparison to be made with the first half.	We have replaced “broad” by “broader”.
76	L107-9: Can directionality of effects be added? Generally strong negative responses to stressors, except for Phosphorus where both groups responded positively, and warming where fish responded positively (and inverts were the only taxonomic group to respond negatively)?	We have modified the sentence as follows: “Only invertebrates consistently displayed strong and negative relations to all stressors except phosphorus enrichment (...) .”

77	L109: With slightly more description added to the first sentence, the additional detail might be needed specifying the contrast, i.e., “In contrast to the microorganism groups, microorganisms...”	We have added: “In contrast to invertebrates, (...)”. Please compare also our response to comment 78.
78	L109-11: I’m not sure this conclusion holds: For taxonomic richness, the variability in responses to stressors is equal for bacteria, macrophytes and fish in terms of 2/7 negative or positive effects. Likewise for evenness, bacteria, algae, macrophytes and fish all have 3/7 positive or negative effects. So really shouldn’t the conclusion be something like; consistency in response to stress is really only observed among invertebrates, which are negatively affected by all stressors except for phosphorus enrichment, thus highlighting their importance as biological indicators of stress/pollution/habitat degradation etc.? I also note that some of this is subsequently picked up in, e.g., paragraph 135-140 on macrophyte responses, where the contrasting positive and negative effects to different stressors is discussed, as well as the paragraph beginning L147 on fish; “Fish exhibit a mix of positive and negative stressor-response relationships.”.	Thank you for spotting this important inconsistency. We have now largely rewritten the paragraph to account for this observation. It reads now: “Only invertebrates consistently displayed strong and negative relations to most stressors except phosphorus enrichment, reflecting their dependence on oxygen availability, habitat structure, and stable flow conditions ²⁵⁻²⁷. In contrast to invertebrates, microorganisms - particularly bacteria/archaea - showed more variable patterns, likely due to their dependence on microscale conditions and the limited availability of suitable datasets ²⁸⁻³⁰. Fish exhibit a mix of positive and negative stressor-response relationships, while relations of macrophytes to oxygen depletion and flow is opposing those of other groups, emphasizing their unique adaptations as rooted, sessile autotrophs (Fig. 2). (...)”
79	L111-113: Again, a conclusion that doesn’t appear to be fully supported by the results – macrophytes showed the same (negative) response to all taxonomic groups for salinity-increase (except bacteria), the same positive response to warming as all groups (except invertebrates), the same response to P-increase re. taxon	We agree that our conclusion was too general and have now modified the text as: “(...) while relations of macrophytes to oxygen depletion and flow is opposing those of other groups, emphasizing their unique adaptations as rooted, sessile autotrophs.”

	richness, though evenness showed the opposite response (could this surprising result be affected by assigning a negative prior set to macrophytes for phosphorus – see further comment below), as was also the case for bacteria. These responses highlight the nuance that this conclusion isn't capturing and could be misleading. Suggest a more specific description where this conclusion holds, such as for oxygen-depletion and flow cessation which align more closely with the discussion of their unique adaptations as rooted, sessile autotrophs.	
80	L116-7: As mentioned above – not just bacteria? Shouldn't this description be; “notably all groups other than invertebrates exhibited a positive relationship with temperature.” This seems a somewhat surprising result and warrants further discussion and exploration of why; one could argue according to this study that climate warming will positively affect freshwater organisms other than invertebrates...	This is another important observation and we agree to this conclusion. While we think this consideration is less suited for this paragraph that deals with bacteria/archaea only, we have now added to the section “Research and management implications”: “For example, warming was positively associated to the diversity of most organism groups, in particular fish, but negatively to invertebrates. At least in parts this observation might simply reflect higher species numbers of warmer, downstream river reaches, which is most obvious for fish, while species richness of invertebrates typically declines downstream.”
81	L141: remove comma after “both,”	Corrected
82	L147: As noted previously, suggest using “responses” rather than “relations”.	Please refer to our response to comments 3 and 68.
83	Fig. 2: The different colours are useful for indicating positive vs negative estimates in a busy plot. Adding some small icons for each taxonomic group and stressor could also be a useful visual tool to aid interpretability.	Thank you for this suggestion. We have added icons for the organism groups.

84	Fig. 3: panel a) x-axis needs to be corrected to Nutrient-N (mg/L).	In fact, the figure legend was wrong. Panel a) refers to bacterial evenness response to temperature. We have corrected the legend.
85 Check	L165-77: Why were these HOPs chosen for deeper discussion? It would be interesting to see the response of invertebrates to temperature, given their contrasting response to all other groups for this stressor. Likewise, a plot comparing the strong positive response of fish to warming would be interesting. Ultimately, a separate panel figure for each key stressor, such as one each for nitrogen, phosphorus, warming and fine sediment, with all taxonomic groups displayed could be quite useful. Perhaps one could be chosen for the main text and the others included in supporting information.	We thank the reviewer for the helpful suggestion. We fully agree that visualizing selected stressor-organism group relationships improves interpretability. In response, we revised our Hypothetical Outcome Plots (HOPs) to focus exclusively on taxonomic richness (compare also comments 86 and 89) and now include:  • Invertebrate richness vs. temperature • Fish richness vs. temperature • Invertebrate richness vs. fine sediment • Fish richness vs. fine sediment These were selected because they illustrate either contrasting responses (e.g., invertebrates vs. fish to temperature) or consistently negative responses (e.g., both groups to fine sediment). We view these HOPs as illustrative examples - intended as a proof-of-concept to support our interpretation of key stressor effects - rather than exhaustive visualizations. We accordingly updated the header of Fig. 3 and the HOP section in the main text.
86 Check	L178-89: Similarly, to the above comment, why these two plots? Fine sediment and flow velocity would be an interesting combination, given their known mechanistic interactive potential. Or temperature with other stressors (e.g. nutrients) given efforts to mitigate climate warming, despite positive effects being observed for most groups...potentially fine sediment and warming would be an interesting PDP for discussion/management implications. I'm not sure that fine sediment and oxygen is so useful, as the discussion in L188-9 "mitigating sediment inputs alone may be insufficient unless accompanied by efforts to maintain adequate oxygen levels" does not have a clear management	We thank the reviewer for this detailed and constructive comment. In response, we revised Fig. 4 to reflect more ecologically relevant and management-relevant stressor combinations. The updated panels now show: (a) Invertebrate richness as a function of oxygen concentration and temperature, (b) Fish richness as a function of flow velocity and fine sediment fraction. These pairings were selected based on both ecological plausibility and clarity of modeled response patterns: The new flow velocity + fine sediment plot (panel b) directly addresses the reviewer's suggestion and captures a mechanistically meaningful relation between two hydromorphological stressors known to affect fish communities through habitat degradation and altered flow regimes. The oxygen + temperature plot for invertebrates (panel a) replaces the earlier sediment vs. oxygen PDP, which showed relatively weak gradients. The new plot aligns better with

	application, given maintaining oxygen levels is difficult in practice and is more likely achieved by reducing nitrogen/phosphorus loading which contributes to exacerbated diurnal DO fluctuations and, potentially, overnight hypoxic conditions. The importance of DO for many invertebrate taxa also doesn't seem to be reflected in the PDP, where one might expect the bottom-left corner to be more pink showing reduced richness under low oxygen (i.e. more like the colour gradient of the nutrient-salinity PDP), perhaps this is because the gradient stops at 5 mg/L and therefore doesn't reflect fully hypoxic conditions – could the gradient be extended? Currently, the concluding sentence could actually be interpreted in the opposite way, given that invertebrate richness is still relatively high (35-40) at the lowest oxygen level shown. I.e. “Our results suggest that mitigating fine sediment inputs could strongly improve invertebrate richness, even under low oxygen conditions, highlighting the pervasive effects of this stressor and the importance of management implications to control its entry into freshwaters” might be a more valid and important conclusion to draw from this PDP.	ecological expectations, showing reduced invertebrate richness under combined additive stress from low oxygen and high temperature. This combination is particularly relevant given increasing concerns about warming-induced oxygen stress in freshwater ecosystems. We also acknowledge the reviewer's concern that the earlier PDP could have led to a misleading interpretation due to the limited range of oxygen values in the data (mostly above 5 mg/L). The revised PDP includes a broader and more representative range of oxygen concentrations and reveals a stronger decline in richness under low-oxygen / high-temperature conditions. Accordingly, we have updated the figure caption and main text to reflect these changes and avoid overinterpretation of management implications where direct mitigation (e.g., of oxygen) is limited.
87	L180-1: Describing the nutrient-salinity PDP in the context of increased salinity offsetting benefits of nitrogen enrichment on algal evenness also seems like an interesting choice description and plot to present among the many options, given most management strategies in freshwater systems seek to mitigate nitrogen loading due to effects described in the	We appreciate the reviewer's thoughtful reflections on the nutrient-salinity PDP and the associated interpretation. We agree that the interaction could be interpreted in multiple ways and also acknowledge the useful language suggestion regarding “counteracts” versus “offsets.” However, in the course of revising the manuscript and responding to prior comments, we have updated Fig. 4 to display more robust and interpretable stressor combinations and

	previous comment (increased algal growth leading to greater diurnal DO cycles and overnight hypoxia). Concerning the description, using the term “counteracts” rather than “offsets” might be better in the context of counteracting benefits of nutrients, whereas offsetting is likely better used when affecting negative effects. Beyond this, however, I would again suggest the gradient of this PDP could be interpreted differently. The slope of this gradient could suggest that nutrient enrichment offsets the negative effects of raised salinity, given the upper-right corner at very high salinity levels is not as pink as might otherwise be expected (evenness is still around the midpoint of the gradient at ~0.3)	focus on richness metrics. As a result, the nutrient-salinity PDP for algal evenness is no longer included in the current version of the manuscript. We now prioritized combinations where stressor-response surfaces were ecologically clearer. We believe the revised figure now offers stronger ecological and management relevance, while still illustrating the conditional nature of stressor impacts on biodiversity. Accordingly, we have re-written the text on the PDPs significantly.
88	L181-3: This sentence might fit better at the end of this paragraph, although I'm not sure it's needed.	We have opted to remove this sentence, also to allow for additional space for additions the reviewers rightly suggested.
89	A general comment on this results section: Possibly choosing either evenness or richness for the HOPs and PDPs would aid interpretability, as it allows the reader to consistently think of the same response when looking across plots. Plots for the other response could be included in Supp infos. I also suggest changing or removing the subheading “Macro- and microorganisms exhibit divergent responses to stressors” as this isn't really that informative, and I'm also not sure it's the most interesting result to highlight.	We thank the reviewer for this helpful suggestion. In line with this recommendation - and to improve consistency and interpretability - we now focus exclusively on taxonomic richness in all HOPs and PDPs presented in the main text. Plots based on evenness have been removed. This decision also reflects limitations in evenness data availability across some organism groups, and the more robust and interpretable richness-based gradients in the selected combinations. We have replaced the subheading by “Relations of biodiversity patterns to single and combined stressor gradients”, which better reflects the contents of the section.
90	I invite the authors to consider a revised results section that is structured around the stressors, rather than the organism groups. This might streamline the	Thank you for this suggestion. During the manuscript development phase, we explored both stressor-structured and organism groups-structured versions of the results and discussion sections. We fully agree that organizing by

	descriptions and reduce the need for some of the contradicting sentences with respect to which taxonomic groups show the most divergent responses to stressors. Of course, different taxonomic groups might have different responses to stress, given their different functional roles at various trophic levels. What is more interesting is how the different stressors manifest effects, at what strength and in which direction across different trophic levels in an ecosystem and which taxonomic groups are more strongly affected (positively or negatively) and why. Thus, a stressor-structured results section would, in my opinion, be more interesting and relevant for management implications that must consider the whole ecosystem, not just separate taxonomic groups. This is the discussion I would expect to read in a “Global scale quantification of stressor responses (in five riverine organism groups)” – the text in parentheses could also be changed to “in riverine ecosystems”, the important part is the quantification of stressor responses.	stressor would better highlight how individual stressors affect different trophic levels and clarify which taxonomic groups are most sensitive. However, organizing the text by organism groups allows clearer comparison of how multiple stressors associate to a given group, which we consider equally important—particularly for practitioners focused on monitoring or managing specific taxa. Following your detailed feedback (comments 76–80), we have significantly revised and streamlined the structure and language of the results section to improve clarity and consistency. Given these revisions, we have chosen to retain the current group-based structure, which we feel now offers a balanced and accessible synthesis across taxa and stressors. We hope the revised text effectively conveys the complexity and variability of stressor–response relationships in riverine ecosystems. We also appreciate the suggestion to revise the title phrasing. However, we think the organism-group focus should somehow be reflected in the title. If we just write “in riverine ecosystems” it will be less obvious what the term “stressor responses” refers to - it could then also be ecosystem functions or abiotic parameters. We therefore would like to keep the phrase “in five riverine organism groups”.
91	L202 (Section; Research and management implications): I think this section would flow better starting with the discussion of the stressor-response relationships relevant to management as discussed in paragraph(s) beginning L225. The paragraph(s) beginning L203 would then fit better toward the end of this section, as they are discussing implications and future directions for the research field, and begin with “Beyond estimating the stressor-response relationships”,	We thank the reviewer for this constructive suggestion to improve the flow and structure of the Research and management implications section. We fully agree that beginning the section with management-relevant findings enhances readability and better aligns with reader expectations - particularly by linking observed stressor–response patterns directly to practical implications for biodiversity conservation. In response, we have reordered the section to first present the key management-relevant results (e.g. stressors with consistent negative effects, context-dependent responses such as warming and nutrient enrichment), followed by

	therefore it would make sense to have already discussed these. If this order is used, the subheading could be re-arranged to “Management and research implications” to reflect the section structure.	methodological and research-related implications (e.g. data availability, reporting practices, and future modeling directions). We have also revised the subheading to “Management and research implications” to better reflect this new structure.
92	L297 – Why did macrophytes get assigned a negative prior set for nutrients-N and P? Would they not respond positively to nutrient addition?	(this comment refers to the supplementary information) Thank you for raising this point. This is likely an artifact of the search for evenness metrics for macrophytes, which are rarely presented in literature. We changed the following: “Nutrient-N and P: Only primary producers are expected to have a relation, where algae got a positive prior set and macrophytes a negative prior set on richness and a positive on coverage (%). The latter positive relation is not displayed in Figure S6.” However, we maintained the negative prior set on the richness of macrophytes, as nutrient loading often leads to the competitive dominance of a few fast-growing, nutrient-tolerant species, resulting in reduced richness.
93	L338-9: It seems surprising that the main model for bacteria and sediment enrichment would have a much more negative logged odds ratio compared to the uninformative model, given that neutral priors were used for bacteria. What is causing this result?	(this comment refers to the supplementary information) We thank the reviewer for raising this important and nuanced point. To clarify this issue, we have extensively expanded the description of this phenomenon (Step 8 in the Supplementary Information), specifically noting how this phenomenon emerges under extreme data sparsity and why it leads to the observed posterior behavior. We also explicitly reference this in the figure descriptions for Fig. S10. We additionally provided two other figures that show the difference between the posterior of an informed model (M1) and a model with only a diffuse prior (M0). Moreover, we would like to emphasize that this pattern can be interpreted within the broader Bayesian epistemological framework: when data are limited, prior knowledge contributes proportionally more to the posterior. This stands

		in contrast to frequentist inference, where bias is typically defined as deviation of the data from a fixed true parameter value over repeated samples. In Bayesian inference, parameters are not fixed but represent distributions of belief, and thus the posterior reflects the current state of information. What may appear as “bias” under a frequentist lens is, in a Bayesian framework, a provisional update of belief given limited ‘evidence’, where bias is difficult to formalize. The text in Step 8 of the Supplementary material now includes the following: “For the sensitivity analysis, the results from the main model (M1) were compared to those from a model using only a diffuse prior $N(0,10)$ for all stressor-response relationships (M0). The natural log of the ratio between M1 and M0 was taken, yielding a log-odds ratio. This ratio quantifies how much prior models contribute to the posterior estimates. A vertical black line indicates no difference between the two models (Fig. S10). For certain stressor-organism relations (e.g., algae and salinity, macrophytes and oxygen, sediment and bacteria, thermal stress and fish), deviations were observed. Notably, prior information often suggested more negative estimated parameters than those inferred by the data only. This trend was particularly evident in taxonomic groups with fewer observations, such as bacteria, algae, and macrophytes. For bacteria and sediment in the logit-linear model, the log-odds ratio is negative even though a neutral prior was used. The diffuse prior as comparison for the sensitivity analysis has only minimal information. Contrary, the neutral prior contains information, but is neutral to the direction of the stressor-response relation (either $N(-0.3, 0.3)$ or $N(0.3, 0.3)$). When data are sparse ($n = 1$ in this case) and lean toward the prior’s structure, the prior exerts stronger influence. This does not reflect a directional bias, but rather the relative weight of prior information under conditions of sparse data. However, as can be observed in Fig. 2 in the main text, the posterior estimate is close to shrunken to 0. This is contrary to the only estimate in the data as -1.47 ($se=0.73$; Fig S2).
--	--	--

		This indicates information in the likelihood has been provided is limited. A similar pattern is seen for macrophytes and oxygen depletion. For fish, the log-odds ratio is marginally positive, indicating that the main model implies a slightly less negative effect than than M0. These patterns can be consistently interpreted across other groups in Fig. S10 in combination with Fig. S2 and Fig. 2. As a general outcome, small sample sizes amplify the influence of prior assumptions on posterior estimates. It is also possible to overlay the density plots of the posterior density distribution of M1 and M0 for both log-linear (Fig. S11) and logit-linear models (Fig. S12). This shows that M0 is often more negative or positive. For the log-linear models: algae vs. oxygen-depletion, bacteria vs. sediment-enrichment, and ish vs. warming. On the other hand, some posteriors for M1 are more concentrated into a specific direction, such as Bacteria vs. flow-cessation, algae vs. warming, macrophytes vs. salinity increase, oxygen depletion, flow-cessation, and N-increase, invertebrates vs. sediment-enrichment and warming, fish vs. salinity-increase, and sediment-enrichment. The same patterns can be observed for the logit-linear models. These trends were explicitly observed for taxonomic groups with fewer observations such as bacteria and macrophyte consistent with the previous check.”
94	L340-1: The trend seems to be most evident in fish which are not listed. Whereas some of the listed groups show neutral or even positive differences between M1 and M0, e.g. for oxygen-depletion, sediment-enrichment for algae and macrophytes and N/P-increase.	(this comment refers to the supplementary information) Based on this comment we have adjusted the SI material for Step 8 expanded and included a more thorough discussion as given in point 93.
95	L352-4: Repeated sentences from an edited version – remove one version.	(this comment refers to the supplementary information) We removed one of the repetitive sentences.

References cited

Badr, Hamada S., Benjamin F. Zaitchik, and Seth D. Guikema. 2014. Application of Statistical Models to the Prediction of Seasonal Rainfall Anomalies over the Sahel. *Journal of Applied Meteorology and Climatology* 53(3): 614–36. doi:10.1175/JAMC-D-13-0181.1.

Greenland, Sander, James J. Schlesselman, and Michael H. Criqui. 1986. The Fallacy of Employing Standardized Regression Coefficients and Correlation as Measures of Effect. *American Journal of Epidemiology* 123(2): 203–8. doi:10.1093/oxfordjournals.aje.a114229.

Penney, Jeffrey. 2017. “A Self-Reference Problem in Test Score Normalization.” *Economics of Education Review* 61: 79–84. doi:10.1016/j.econedurev.2017.10.003.

Shiroyama, Risa, and Chihiro Yoshimura. 2016. Assessing Bluegill (*Lepomis Macrochirus*) Habitat Suitability Using Partial Dependence Function Combined with Classification Approaches. *Ecological Informatics* 35: 9–18. doi:10.1016/j.ecoinf.2016.06.005.

Reviewer 1.		
Nr.	Comment	Response
1	I appreciate the additions to the SI about the data included. You've now well-described the aggregate dataset. But I am missing something for the main text. I expect to see a sentence something like this: "Each study sampled a riverine community at least 5 times (mean = X), across at least X sites (mean=X) in X years (mean =X), focusing on at least one stressor (mean = X stressors)". This could go somewhere around line 88.	Thank you for this suggestion. We have added the following sentence to the main text: "Each study sampled a riverine community at least six times (median = 14, mean = 58, SD = 345), focusing on at least one stressor (median = 3, mean = 3.3, SD = 2.5)." Details on sites and years could not be reported consistently, as many studies did not clearly distinguish spatial from temporal gradients. We therefore modelled random factors for temporal and spatial structures whenever available and now note this limitation in the Discussion (see comment 2 below).
2	Apologies if my last comment was not clear enough, but its still not clear whether your datasets study temporal or spatial gradients in stressors. E.g., is the x-axis for the stressor effect within a dataset (step 1) representing different sites/ivers (each with different stressor values) within the same year, or different years (with different stressor values) at the same sites. Its well-known that different strengths of relationships can be found since different ecological processes affect each (with lags especially affecting temporal effects, and confounding especially affecting spatial effects - hence the whole space-for-time substitution debate). A paper unpacks it here: Birds and Climate in Space and Time: Separating Spatial and Temporal Effects of Climate Change on Wildlife –	Thank you for raising this important point. Our dataset primarily comprises studies that capture spatial gradients (sites or rivers with varying stressor levels sampled within a given period), although some studies also include temporal gradients. Unfortunately, many publications do not explicitly provide details on the gradient. In our meta-analysis, we incorporated random effects for both temporal (e.g. year, season) and spatial (e.g. site, location) grouping structures when available. The meta-analysis was conducted on elasticity or semi-elasticity coefficients derived from log–log or logit–log models of the form: $g(y) = \beta_0 + \sum_{i=1}^n \beta_i \cdot \log(x_i) + t_i + s_i$ where g is the log or logit link, t represents random temporal effects (e.g., year, season, month), and s represents random spatial effects (e.g., site, location). These random effects were included during

	Methods Blog I appreciate your paper is not delving into such detail and I am not suggesting you follow exactly the methods of this paper. But it still needs to be clear whether the studies focus on spatial or temporal stressor variation. This is linked to my comment above about the summary of the data. Based on your current inclusion criteria (at least 5 obs – without specifying the units of those obs across space or time), there is the impression that you have both temporal and spatial studies; though I am guessing probably mostly the latter. Spatial gradients are typically easier to study (within a typical short-term project), but for many questions e.g., making predictions about future change, we want to make inferences about temporal changes (again, linking with the space-for-time issue). If you do have a mix of studies, then this could explain some variation in your results, and so ideally would have been tested or controlled for (by labelling each study as ‘spatial’ or ‘temporal’). If this is not possible, this is a limitation that could be briefly mentioned in your Discussion, and unpacked in future work.	model fitting (step 2). However, two main limitations constrained this approach: (i) not all studies provided explicit spatial or temporal gradients, and (ii) the number of observations per study was often small (median = 15). Attempts to model both gradient structures simultaneously under these conditions would quickly lead to overparameterization and convergence issues due to limited degrees of freedom. We now explicitly acknowledge this limitation in the Discussion: “In addition, small sample sizes and incomplete reporting constrained our ability to account for spatial and temporal residual autocorrelation, likely contributing to the heterogeneity among stressor-response relationships.” We also (already) emphasize that our Bayesian framework allows sequential updating, so that as more detailed datasets become available, the present results can be refined accordingly.
	Line 120-121 – is bold needed here?	No; thank you for pointing this out, we adjusted this accordingly.
	Line 88 – ‘uncertainty’ instead of ‘clarity’	Thanks for the suggestion, we changed it in the text accordingly.

Reviewer 2		
Nr.	Comment	Response
	In general, an improved version which has incorporated caveats over points made on the initial submission. I've concentrated mostly on editorial issues that could be addressed to help readability.	Thank you for the positive feedback and the additional input to further refine the text.
	Line 37: 'taxa richness' is not grammatically correct despite being in wide use in the literature - particularly from N America. Taxa are plural, taxon is similar. You mean taxon richness.	Thank you for pointing this out. We corrected this accordingly.
	Line 37. Relation with, not relation of	We accordingly replaced 'relation of' to 'relation with'
	Line 26: shouldn't this say that reduced biodiversity (...) reflected elevated salinity, oxygen depletion, and fine sediment accumulation? As written, the implication is that biodiversity affected physicochemical conditions rather than the reverse	We now changed the sentence to: "Consistently, biodiversity (taxon richness, evenness) across taxa reflected elevated salinity, oxygen depletion, and fine sediment accumulation, while the relation with nutrient enrichment and warming varied among groups."
	Lines 45-47 Here and in other places, the writing could be simpler. "Over recent decades, freshwater ecosystems - particularly rivers - have experienced rapid biodiversity decline caused by multiple, interacting stressors at local, catchment and global scales"	Thank you for the suggestion. We agree that some sentences could be simplified and revised accordingly. For the opening of the main text, we also aimed for greater clarity. However, we opted not to begin with "Over recent decades" because the following paragraph already starts in a similar way. Instead, we revised the sentence to: "Freshwater ecosystems—particularly rivers—have undergone rapid biodiversity declines in recent

		decades, driven by multiple interacting stressors across local, catchment, and global scales.”
	Line 50: agricultural intensification?	Thank you for the suggestion, we happily adopted it.
	Line 68-71: could be simplified	Thank you for highlighting it. We simplified it to: “Although numerous studies have examined individual stressor–response relationships and one global synthesis compared terrestrial, marine, and freshwater communities, a quantitative assessment across aquatic organism groups and stressors is still lacking.”
	Line 120-122: something wrong here	Thanks for pointing that out - we corrected it.
	Line 126: I'd use the past tense to maintain consistency with previous sentences	Thanks for pointing this out - we corrected it accordingly.
	Line 129-130: Perhaps: 'However, our meta-analysis necessarily obscures divergent patterns of macrophytic taxa, such as bryophytes and vascular plants, which vary substantially in evolutionary origins, traits and environmental requirements'.	We adapted the text according to this suggestion.
	Line 139: are or were? Past tense? You have some tendency to switch between tenses.	Thank you for pointing this out. We have revised the manuscript to consistently use past tense throughout the text.
	Line 147: I think this sentence structure mixes dependent and independent variables - here and elsewhere	Thank you for pointing this out. We changed it to “Algal richness and evenness were positively associated with nutrient enrichment, particularly nitrogen, reflecting

		enhanced productivity ” accordingly.
	Line 187-200: past tense for results?	We amended it accordingly.
	Line 225: were?	We changed it accordingly

Reviewer 3		
Nr.	Comment	Response
1	I have re-read this manuscript and reply letter and I am satisfied that the authors have appropriately revised the manuscript, and sufficiently addressed all of my comments and suggested revisions.	Thank you!